



# Implementation of a Satellite-Based Tool for the Quantification of CH₄ emissions over Europe (AUMIA v1.0) – Part 1: Forward Modeling Evaluation against Near-Surface and Satellite Data

Angel Vara-Vela[1,2,3], Christoffer Karoff[1,2,3], Noelia Rojas Benavente[4], Janaina Nascimento[5,6]

[1]Department of Geoscience, Aarhus University, 8000 Aarhus, Denmark
[2]Department of Physics and Astronomy, Aarhus University, 8000 Aarhus, Denmark
[3]iCLIMATE Aarhus University Interdisciplinary Centre for Climate Change, Aarhus University, 4000 Roskilde, Denmark
[4]Department of Atmospheric Sciences, Institute of Astronomy, Geophysics and Atmospheric Sciences, University of São Paulo, São Paulo, Brazil
[5]NOAA ESRL Global Systems Laboratory, Boulder, United States
[6]Cooperative Institute for Research in Environmental Sciences, University of Colorado, Boulder, United States

*Correspondence to*: Angel Vara-Vela (angel@geo.au.dk)

**Abstract.** Methane is the second most important greenhouse gas after carbon dioxide, and accounts for around 10 % of total European Union greenhouse gases emissions. Given that the atmospheric methane budget over a region depends on its

terrestrial and aquatic methane sources, inverse modeling techniques appear as a powerful tools for identifying critical areas that can later be submitted to emission mitigation strategies. In this regard, an inverse modeling system of methane emissions for Europe is being implemented based on the Weather Research and Forecasting (WRF) model: the Aarhus University Methane Inversion Algorithm (AUMIA) v1.0. The forward modeling component of AUMIA consists of the WRF model coupled to a multipurpose global database of methane anthropogenic emissions. To assure transport consistency during the

inversion process, the backward modeling component will be based on the WRF model coupled to a lagrangian particle dispersion module. A description of the modeling tools, input data sets and one-year forward modeling evaluation from April 01, 2018 to March 31, 2019 is provided in this paper. The a posteriori methane emission estimates, including a more focused inverse modeling for Denmark, will be provided in a second paper. A good general agreement is found between the modeling results and observations based on the TROPOspheric Monitoring Instrument (TROPOMI) onboard the Sentinel-5

Precursor satellite. Model-observation discrepancies for summer peak season are in line with previous studies conducted over urban areas in central Europe, with relative differences between simulated concentrations and observational data in this study ranging from 1 to 2%. Domain-wide correlation coefficients and root-mean-square-errors for summer months ranged from 0.4 to 0.5 and from 27 to 30 ppb, respectively. For winter months, otherwise, model-observation discrepancies show a significant overestimation of anthropogenic emissions over the study region, with relative differences ranging from 2 to 3%.

Domain-wide correlation coefficients and root-mean-square-errors in this case ranged from 0.1 to 0.4 and from 33 to 50 ppb, respectively, indicating that a more refined inverse analysis assessment will be required for this season. According to modeling results, the methane enhancement above the background concentrations came almost entirely from anthropogenic





sources; however, these sources contributed with only up to 2 % to the methane total column concentration. Contributions from natural sources (wetlands and termites) and biomass burning were not relevant during the study period. The results

found in this study contribute with a new model evaluation of methane concentrations over Europe, and demonstrate a huge and under explored potential for methane inverse modeling using improved TROPOMI products in large-scale applications.

## 1 Introduction

Atmospheric methane (CH$_4$) has more than doubled since the pre-industrial era. Although it remains in the atmosphere for a

relatively short period of time (~10 years) compared to carbon dioxide (centuries to millennia), its constant emission all over the world makes it a well-mixed greenhouse gas (IPCC, 2021). CH$_4$ concentrations have a direct influence on the climate, but also have a number of indirect effects on human health and vegetation, including crop production (Mar et al., 2022). After decades of steady growth, reaching even a growth rate of approximately zero from 2000 to 2006, the atmospheric CH$_4$ has returned to values observed in the second half of the twentieth century, and in recent years it has increased at a faster rate

(Rigby et al., 2008; Nisbet et al., 2016; Palmer et al., 2021). According to Van Dingenen et al. (2018), if unabated, the global anthropogenic CH$_4$ emissions could increase up to 100 % by 2050, thus leading to a general situation in which ozone-related premature mortality and crop damage events linked to CH$_4$ emissions would be more frequent. In the European Union (EU), 53% of anthropogenic CH$_4$ emissions come from agriculture, 26% from waste and 19% from energy, with these sectors accounting for up to 95% of CH$_4$ emissions associated with human activity worldwide (European Commission, 2020).

Improving the quality of CH$_4$ emissions data for these concerned key sectors in the EU inventory has been mandatory in recent years (EEA, 2022), with the implementation of emissions monitoring technologies, including satellite missions such as the TROPOspheric Monitoring Instrument (TROPOMI) onboard the Sentinel-5 Precursor satellite. In addition, some major initiatives involving the use of atmospheric inverse modeling at the global scale, with emphasis given to greenhouse gases that have large anthropogenic sources, have been implemented in order to respond to an identified demand from the

climate community at large (Bergamaschi et al., 2018).

Prior to an inverse analysis as such, a robust evaluation using chemical transport models and satellite observations is usually performed to identify and quantify deficiencies in the CH$_4$ emission model. Such comparative studies have focused mostly on CH$_4$ column-averaged dry air mole fractions (hereafter referred to as XCH$_4$ concentrations) from the SCanning Imaging Absorption spectroMeter for Atmospheric ChartographY (SCIAMACHY) and Thermal And Near-infrared Sensor

for carbon Observation (TANSO) instruments onboard the Environmental Satellite (EnviSat) and Greenhouse gases Observing SATellite (GOSAT), respectively. However, since it was made publicly available to the community in April 2018, TROPOMI XCH$_4$ data have been exploited in numerous studies not only for validation purposes (e.g., Zhao et al., 2019; Zhao et al., 2022; Callewaert et al., 2022) but mainly to optimize emission estimates (e.g., Varon et al., 2022; Chen et al., 2022). TANSO provides more mature but sparser XCH$_4$ concentrations than TROPOMI, and is together with TROPOMI

the only two satellite instruments that remain operational since they were launched in 2009 and 2017, respectively. Qu et al. (2021), in a one-year global validation of TROPOMI and TANSO XCH$_4$ retrievals with Total Carbon Column Observing



Network (TCCON) $CH_4$ total column measurements, have shown larger biases with TROPOMI in some regions of the world. Nevertheless, with further improvements in the retrieval algorithms, e.g. as those implemented by Lorente et al. (2021) to correct systematic biases in low- and high-albedo regions, TROPOMI high observation density and resolution will likely

improve forward modeling evaluation and inversion results (Hu et al., 2018; Jacob et al., 2022).

In addition to a chemical transport model to relate emissions to satellite observations, Bayesian inversion techniques require a cost function to fit the satellite observations to the model predictions, and a priori estimates of emissions to regularize the solution where the observations provide insufficient information (Brasseur and Jacob, 2017). Inverse modeling studies of $CH_4$ emissions available for Europe have been mostly performed at global scale and based on TANSO

observations (e.g., Tsuruta et al., 2017; Segers et al., 2020), with just a few based on TROPOMI such as the Integrated Methane Inversion (IMI) v1.0, a cloud-based facility developed to support a growing demand for tools to infer regional $CH_4$ emissions (Varon et al., 2022). IMI v1.0 exploits the GEOS-Chem chemical transport model and its nested capability to simulate $CH_4$ concentrations over inversion domains at 0.25°×0.3125° resolution, with dynamic boundary conditions from a global archive of smoothed TROPOMI data. In 2014, the EU's Earth Observation Programme implemented the Copernicus

Atmosphere Monitoring Service (CAMS) for developing information services based on environmental monitoring satellites. CAMS global inversion-optimised $CH_4$ fluxes are constrained based on TANSO measurements, and are available for the period 1990-2020 at a 2°×3° resolution (Segers et al., 2020). Inverse modeling studies using in-situ (e.g., Bergamaschi et al., 2018) and ground-based total column (e.g., Wunch et al., 2019) measurements instead of satellite observations have also been conducted – inversions yielded, depending on the European region, higher/lower $CH_4$ emissions with regard to the

EDGAR-based a priori emission estimates. A review of the global $CH_4$ budget by Saunois et al. (2019) found significant discrepancies between $CH_4$ emission estimates using bottom-up and top-down approaches, with most of the discrepancies being attributed to uncertainties in natural sources. Recent inverse modeling studies combining $CH_4$ concentrations with isotopic signature of $CH_4$ ($\delta^{13}C$-$CH_4$) attribute roughly 85 % of the post-2006 growth in atmospheric $CH_4$ to microbial sources, with about 50 % coming from the tropics (Basu et al., 2022).

A few studies have combined model simulations with satellite observations to characterize $CH_4$ concentrations over Europe. Hence, this study aims at evaluating recent improvements to atmospheric modeling tools and satellite measurements made by the atmospheric modeling community. This is the first in a serial of two papers that aim to implement an inversion system of $CH_4$ emissions for Eurpope based on the next-generation TROPOMI $XCH_4$ measurements: the Aarhus University Methane Inversion Algorithm (AUMIA) v1.0. Here, we evaluate $XCH_4$ concentrations derived from the AUMIA forward

modeling component coupled to a multipurpose global database of $CH_4$ anthropogenic emissions, against the Netherlands Institute for Space Research (SRON) S5P-RemoTeC $XCH_4$ product version 17. This is a new scientific TROPOMI $XCH_4$ product that presents substantial improvements in relation to the operational product. Several two-week periods in 2018 and 2019 were carefully selected for model sensitivity tests. Then, one-year simulation period from April 01, 2018 to March 31, 2019 was performed for model validation. In addition, simulated $CH_4$ concentrations have been compared to near-surface

observations from the Integrated Carbon Observation System (ICOS) network. In the second part of this work, we will





provide a posteriori CH$_4$ emission estimates based on the WRF model coupled to a lagrangian particle dispersion module which is currently under development. It will be also provided model evaluation and inverse modeling of CH$_4$ emissions for Denmark. The paper is arranged as follows. In section 2, the CH$_4$ observations and modeling tools, including a description of the experimental design, are introduced. Next, in section 3, the forward modeling performance is evaluated by comparing the
model results against near-surface and total column observations. Section 4 will discuss the contributions of anthropogenic sources to the XCH$_4$ concentration. Finally, a summary and concluding remarks are given in section 5.

## 2 Data and Methods

### 2.1 WRF-GHG model

The core component of AUMIA v1.0 is the Weather Research and Forecasting (WRF) model (Skamarock et al., 2021). WRF
is a fully compressible, non-hydrostatic model supported by the National Center for Atmospheric Research (NCAR) to a worldwide community of users. Due to its robustness and versatility, WRF has been widely used for operational forecasts and research related to severe weather and air pollution (e.g., Vara-Vela et al., 2021), the latter through the use of its chemistry extension, the WRF-Chem model (Grell et al., 2005). The WRF Greenhouse Gas model (Beck et al., 2011), hereafter referred to as WRF-GHG, is selected as the forward modeling component of AUMIA. WRF-GHG is an extension
to the WRF-VPRM model (Ahmadov et al., 2007) which couples the WRF model to the Vegetation Photosynthesis and Respiration Model (VPRM) (Mahadevan et al., 2007). WRF-GHG simulates CH$_4$ concentration based on emission estimates from external data sets as well as from online calculations driven by model parameters such as soil moisture, soil temperature and vegetation type. CH$_4$ fluxes from external data sets, specifically for anthropogenic (except for biomass burning) and biomass burning sources, are converted into atmospheric concentrations based on flux models. On the other
hand, online calculations comprise CH$_4$ emissions from wetlands and termites, and CH$_4$ uptake by soil. CH$_4$ contributions from anthropogenic, biogenic (wetlands, termites and soil uptake) and biomass burning sources, as well as those from background concentrations are separately determined using tagged tracers. WRF-GHG allows for passive transport (i.e., without any chemical loss or production) of not only CH$_4$, but also of carbon dioxide and carbon monoxide which undergo advection and convective mixing as any other chemical species. WRF-GHG was incorporated into the WRF-Chem model
for the first time at its version 3.4, and is since then one of the many available chemistry options in this model. A detailed description of the WRF-GHG model, its emission preprocessors and related modules can be found in Beck et al. (2011) and Beck (2012). In this work, WRF-GHG was run as a chemistry option in the WRF-Chem model version 4.3. Implementing the AUMIA v1.0 will enable us to extend its application to other greenhouse gases such as carbon dioxide, e.g., by incorporating new satellite missions such as the Copernicus Anthropogenic Carbon Dioxide Monitoring (CO2M).




### 2.1.1 Grid configuration

The experimental setup consisted of two nested domains configured in a Lambert conformal projection at horizontal resolutions of 30 and 10 km. The parent domain has 120×120 grid points and is defined to cover most of Europe, whereas the nested domain (D02) has 67×61 grid points and focuses on Denmark (see Figure 1). The 10 km grid spacing domain
covering Denmark is motivated by improving the country greenhouse gases quantification. WRF-GHG uses an Arakawa C-grid staggering and a hybrid vertical coordinate which is a coordinate that is terrain-following near the ground and becomes isobaric higher up. The vertical resolution includes 45 layers extending from the surface up to 1 hPa, with more closely spaced layers at lower altitudes. Static geographical data (e.g., topography, land use) and masked surface fields are derived from Moderate Resolution Imaging Spectroradiometer (MODIS) and U. S. Geological Survey (USGS) products. Tables 1
lists the main grid configuration features used in the simulations.

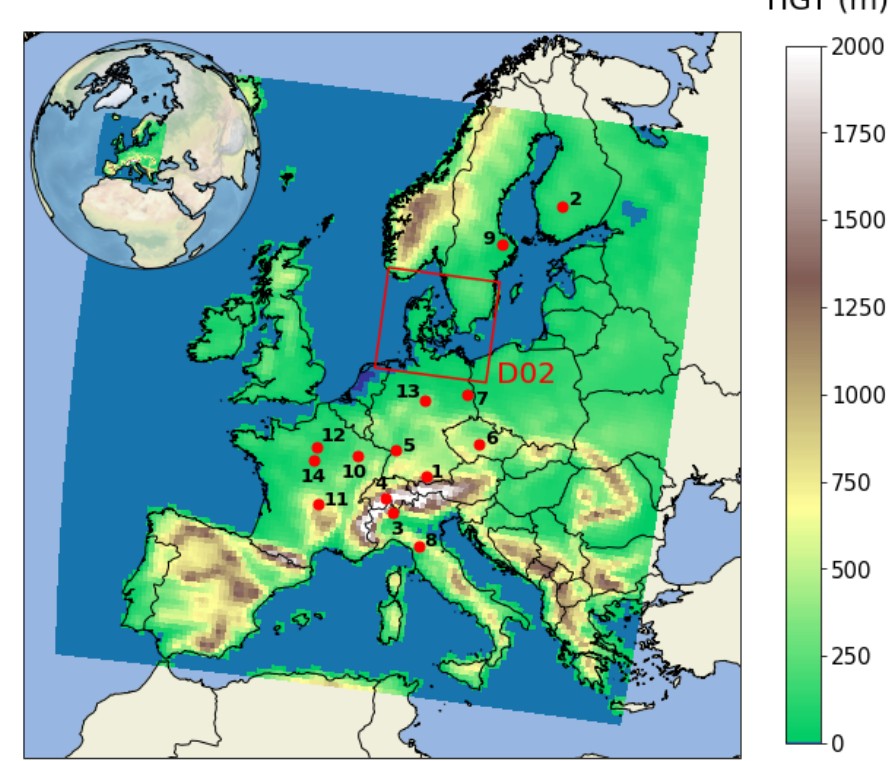

**Figure 1.** Modeling domains. The parent domain covers most of Europe, whereas the nested domain (D02) focuses on Denmark. The red markers, numbered from 1 to 14, indicate the location of the ICOS stations considered for model evaluation. ICOS station features are presented in detail in Table 3.





**Table 1.** WRF-GHG grid configuration.

| Attributes | Model parameter/coverage |
| --- | --- |
| Domains | 120×120 (latitude×longitude) grid points over Europe, 67×61 (latitude×longitude) grid points over Denmark |
| Center-point of the parent domain | 51.98 °N, 5.66 °E |
| Map projection | Lambert conformal |
| Horizontal and vertical resolution | 30 and 10 km – 45 sigma-type levels |
| Model top | 1 hPa (~44 km) |
| Time step | 180 and 60 s |
| Static data | Topography: USGS, 30 s resolution<br>Land use: MODIS, 21 land use categories |
| Grid relaxation zone | 5 points |

## 2.2 Input emissions

### 2.2.1 Anthropogenic fluxes

Anthropogenic fluxes of $CH_4$ (not including biomass burning sources) are externally prepared based on the Emissions Database for Global Atmospheric Research (EDGAR) version 6 Greenhouse Gas Emissions (Crippa et al., 2021). EDGAR has been widely used in support of policy design for greenhouse gases emissions verification, using international statistics and a consistent Intergovernmental Panel on Climate Change (IPCC) methodology. Statistical information compiled by the IPCC Guidelines for National Greenhouse Gas Inventories (IPCC, 2006) is adopted by EDGAR for most sources and

countries, complemented with information from scientific literature and other references for specific processes and/or countries. A detailed description of data providers and technical procedures for the greenhouse gases emissions of EDGAR can be found in Janssens-Maenhout et al. (2019). EDGARv6.0 includes a set of key novelties such as country/region- and sector-specific yearly profiles for all sources and country-specific weekly and daily profiles to represent hourly emissions. EDGARv6.0 $CH_4$ fluxes in this work include activity data from 24 different sectors that can be grouped into the following

broad sectors: energy, industry, aviation, ground transport, shipping, agriculture and waste. Biomass burning fluxes from human activities were prepared separately using a satellite-based emissions preprocessor (Wiedinmyer et al., 2023). EDGARv6.0 $CH_4$ fluxes are provided as monthly grid maps spatially distributed on a common grid at 0.1°×0.1° resolution, and can be freely downloaded at http://jeodpp.jrc.ec.europa.eu/ftp/jrc-opendata/EDGAR/datasets/v60_GHG/. Figure 2 shows the spatial distributions of $CH_4$ emission rates for different sectors for May 2018 in the 30 km modeling domain.






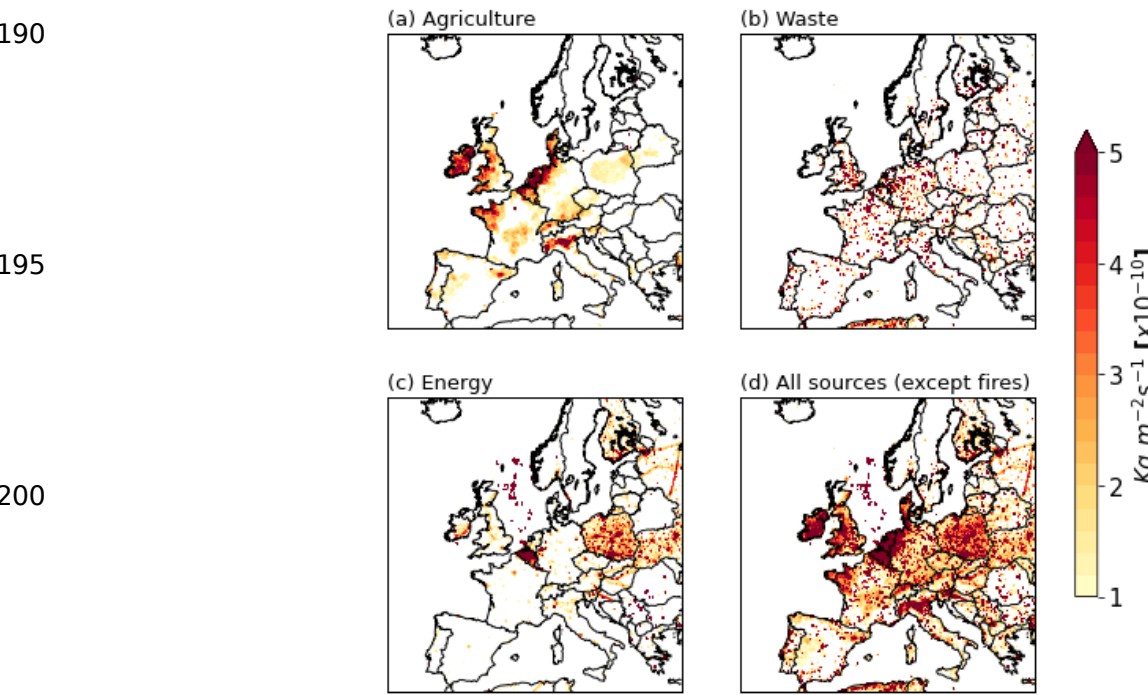

**Figure 2.** Spatial distribution of EDGARv6.0 CH$_4$ emission rates for the concerned key sectors for May 2018 in the 30 km modeling domain. Each grid point in panel (d) represents the sum of emission rates from all different sectors (except biomass burning) based on country-specific temporal profiles.

### 2.2.2 Biogenic fluxes

Anaerobic microbial production of CH$_4$ in wetlands represents the dominant source of CH$_4$ emissions from nature, followed by CH$_4$ emissions from termites. Uptake of atmospheric CH$_4$ by soil is the only terrestrial sink. CH$_4$ fluxes from natural source and sink processes are all calculated online in the model simulations. CH$_4$ fluxes from wetlands are determined as a percentage of the heterotrophic respiration (Christensen et al., 1996) using the approach of Sitch et al. (2003) and the WRF-GHG variables soil moisture and soil temperature. A wetland inundation map (Kaplan et al., 2002) is then applied for the

determination of the wetland fraction per grid cell. CH$_4$ fluxes from termites are calculated, based on the global data base for termite emissions described in Sanderson (1996), as the product of the biomass of a population of termites and the flux of CH$_4$ emitted from that termite population. A mapping of the vegetation types used by Sanderson (1996) to the WRF-GHG vegetation type is previously performed for the quantification of the termite's biomass per grid cell. Based on WRF-GHG driving variables such as soil moisture, soil temperature, CH$_4$ concentration and precipitation, soil uptake fluxes are





calculated following the approach devised by Ridgwell et al. (1999). For wetland grid cells (i.e., grid cells dominated by wetlands), the calculation of soil uptake is suppressed as this process does not take place over flooded areas.

### 2.2.3 Biomass burning fluxes

Biomass burning fluxes of $CH_4$ are externally prepared based on the Fire INventory from NCAR version 2.5 (FINNv2.5) (Wiedinmyer et al., 2023). FINNv2.5 uses satellite observations of active fires and land cover, together with emission factors
and fuel loadings to provide daily, highly-resolved (1 km) open burning emissions estimates for use in chemical transport models. Active fire products from both MODIS instruments onboard the Terra and Aqua satellites are applied, and to avoid double counting of the same fire on a single day, multiple detections of the fire in question are identified globally and then removed as described by Al-Saadi et al. (2008). While $CH_4$ fluxes from anthropogenic and biogenic sources are added at the first model level, a plume rise algorithm is applied to determine the injection height of biomass burning plumes. The plume
rise algorithm, implemented in the WRF-Chem by Grell et al. (2011), is based on the 1-D time-dependent cloud model developed by Freitas et al. (2007). The algorithm is called for numerical integration for grid cells that contain fire spots, with the lower and upper limits of the injection height being calculated based on the fire category (biome burned) provided by the FINNv2.5, as well as heat flux fields inferred from WRF-GHG.

### 2.3 Experiment design

Initially, a model sensitivity analysis for evaluation of model parameterizations such as planetary boundary layer and cumulus clouds, as well as global forcings for $CH_4$ concentration, was carried out over several two-week periods in 2018 and 2019. Then, based on the model configuration that best fit the satellite data, one-year simulation period from April 01, 2018 to March 31, 2019 was performed. Initial and boundary conditions to drive the simulations at 30 km are based on global data from the European Centre for Medium-Range Weather Forecasts (ECMWF) Reanalysis v5 (ERA5) model (Hersbach et al.,
2020), for meteorological processes, and from the NCAR Community Atmosphere Model with Chemistry (CAM-chem) (Lamarque et al., 2012; Emmons, et al., 2020), for background concentrations of $CH_4$; both called for input every 6 h. Off-line initial and boundary conditions derived from the simulations at 30 km are used as input to feed the simulations at 10 km. Model results and discussion for the nested domain are under development and will be described in a forthcoming paper. The main physical parameterizations included the Rapid Radiative Transfer Model (RRTM) for longwave radiation; the
Pennsylvania State/NCAR Mesoscale Model version 5 (MM5) for shortwave radiation; the Revised Mesoscale Model version 5 Monin–Obukhov scheme for the surface layer (Jimenez et al., 2012); the Unified Noah land-surface model for land surface (Chen and Dudhia, 2001); the Yonsei University scheme for the planetary boundary layer (Hong et al., 2006); the Kain–Fritsch scheme for cumulus clouds (Kain 2004); and the WRF Single-Moment 5-class (WSM5) scheme for microphysics (Hong et al., 2004). Tables 2 lists the physics and emissions schemes used in the simulations. A schematic of
the model running process is depicted in Appendix A.



### 2.3.1 Postprocessing

In order to compare the simulated $XCH_4$ concentrations with the observations, a set of model data posprocessing steps involving a priori information from the satellite retrievals were carried out as follows: (i) satellite information for each orbit was regridded to the WRF-GHG discretization; (ii) simulated concentrations were resampled to the SRON S5P-RemoTeC standard twelve-levels pressure grid; (iii) smoothed concentrations corresponding to the resampled profiles were calculated according to the following linear transformation:

$$CH_{4,smooth} = K \cdot CH_{4,tot} + (I - K) \cdot A \tag{1}$$

where $CH_{4,smooth}$ represents the smoothed $CH_4$ concentration, $A$ and $K$ are the a priori profile and averaging kernel of the retrieval, respectively, $I$ is the identity matrix, and $CH_{4,tot}$ is the total $CH_4$ concentration. $CH_{4,tot}$ is obtained by adding up the tracer contributions from the emission sources and bacground concentrations are:

$$CH_{4,tot} = CH_{4,ant} + CH_{4,bio} + CH_{4,bbu} + CH_{4,bgd} \tag{2}$$

where $CH_{4,ant}$, $CH_{4,bio}$, $CH_{4,bbu}$ and $CH_{4,bgd}$ a $\Delta M_0$ represent the $CH_4$ concentrations from anthropogenic sources, biogenic sources, biomass burning and background concentrations; (iv) the $XCH_4$ concentration was finally calculated as the pressure-weighted concentration following Zhao et al. (2019):

$$XCH_4 = \sum_i \left[ \frac{P_{bottom} - P_{top}}{P_{sfc} - P_{top}} \right] \times CH_{4,smooth} \tag{3}$$

where $P_{bottom}$ and $P_{top}$ represent the pressures at the bottom and at the top of the $i^{th}$ vertical grid cell, and $P_{top}$ and $P_{sfc}$ represent the hydrostatic pressures at the top and at the surface of the model domain, respectively. Simulated total column concentrations without taking into account the a priori information and averaging kernels were also computed to evaluate smoothing effects. In this case, the Equations (2) and (3) are directly applied to the model outputs without any previous smoothing. Model evaluation against in-situ $CH_4$ measurements is performed on the basis of the closest model grid points to the ICOS stations. Three groups of eight, six and five ICOS stations, with sampling heights between 8.0–16.8 m, 40–50 m, and 100 m, respectively, were selected for comparison with simulated $CH_4$ concentrations interpolated to roughly 10, 50 and 100 m above ground level.





**Table 2.** WRF-GHG simulation design.

| Atmospheric process | Scheme/Model |
|---|---|
| Cloud microphysics | WSM5 (Hong et al., 2004) |
| Longwave radiation | RRTM (Mlawer et al., 1997) |
| Shortwave radiation | MM5 (Dudhia 1989) |
| Boundary layer | YSU (Hong et al., 2006) |
| Land surface | Unified Noah land-surface model (Chen and Dudhia, 2001) |
| Surface layer | Revised MM5 Monin–Obukhov (Jimenez et al., 2012) |
| Cumulus clouds | Kain-Fritsch (new Eta) (Kain 2004) |
| Anthropogenic emissions[1] | EDGARv6.0 (Crippa et al., 2021) |
| Wetland emissions | Kaplan (2002) |
| Termite emissions | Sanderson (1996) |
| Soil uptake fluxes | Ridgwell et al. (1999) |
| Biomass burning emissions[1] | FINNv2.5 (Wiedinmyer et al., 2023) coupled to a plume rise module |
| Initial and boundary conditions[2] | ERA5 (0.25°, 37 pressure levels) for meteorology and CAM-chem (0.9°×1.25°, 56 vertical levels) for $CH_4$ concentrations. |
| Simulation period | April 01, 2018 to March 31, 2019 |

[1]The emission files for anthropogenic and biomass burning sources were processed for model input using the NCAR utilities anthro_emis and fire_emis, respectively.

[2]The initial and boundary conditions for $CH_4$ concentrations were prepared using the NCAR utility mozbc.

**2.4 Observational data**

**2.4.1 TROPOMI**

The TROPOspheric Monitoring Instrument (TROPOMI) onboard the Copernicus Sentinel-5 Precursor (S5P) satellite is a spectrometer that provides global coverage of total column concentrations for different gases at an unprecedent resolution of 5.5×7 km$^2$ utilizing a push-broom configuration. The TROPOMI-based $XCH_4$ concentrations in this study are taken from the Netherlands Institute for Space Research (SRON) S5P-RemoTeC $XCH_4$ product version 17, available at https://ftp.sron.nl/open-access-data-2/TROPOMI/tropomi/ch4/. In relation to the operational data products (Hu et al., 2016), the SRON S5P-RemoTeC $XCH_4$ product v17 (Lorente et al., 2022) provides updates regarding the regularization scheme, the selection of the spectroscopic database, the implementation of a higher resolution digital elevation map for surface altitude, and a more sophisticated a posteriori correction for the albedo dependence. The main update with respect to the previous version, the SRON S5P-RemoTeC $XCH_4$ v14 (Lorente et al., 2021), includes $XCH_4$ retrievals over ocean for observations made under sun-glint geometries. A quality data assessment was performed using TCCON and TANSO measurements. The TROPOMI $XCH_4$ data of interest to this work correspond to the recommended high-quality retrievals, with quality assurance value of 1 and for S5P orbits over Europe, i.e., with crossing times between 09:00 and 13:00 UTC.





### 2.4.2 ICOS

The Integrated Carbon Observation System (ICOS) is a pan-European Research Infrastructure that provides harmonized, high-precision, and long-term monitoring of atmospheric greenhouse gases. It sustains a network of stations that spread out over different ecosystems across 12 European countries (Heiskanen et al., 2022). Greenhouse gases concentrations and meteorological parameters are usually taken at different heights of measurement towers set up in mountainous terrain or in remote environments. ICOS $CH_4$ concentrations in this study correspond to the fully quality checked Level 2 data, available

for download at the ICOS Carbon Portal (https://data.icos-cp.eu). For users interested in using ICOS data, we strongly recommend to use the ICOS Carbon Portal pylib, a python library that provides easy access to data hosted at the ICOS Carbon Portal. The ICOS stations used in this study are compiled in Table 3 and their locations are shown in Figure 1.

**Table 3.** ICOS stations and atmospheric parameters considered for model evaluation.


| Station name | Country | Latitude | Longitude | Altitude | Sampling height | Parameters |
|---|---|---|---|---|---|---|
| 1. Hohenpeissenberg | Germany | 47.80 °N | 11.02 °E | 934 m | 50 m | $CH_4$ |
| 2. Hyytiälä | Finland | 61.84 °N | 24.29 °E | 181 m | 16.8 m | $CH_4$ |
| 3. Ispra | Italy | 45.81 °N | 8.63 °E | 210 m | 40 and 100 m | $CH_4$ |
| 4. Jungfraujoch | Switzerland | 46.54 °N | 7.98 °E | 3580 m | 10 m | WS and WD |
| 5. Karlsruhe | Germany | 49.09 °N | 8.42 °E | 110 m | 100 m | $CH_4$ |
| 6. Křešín u Pacova | Czech Republic | 49.57 °N | 15.08 °E | 534 m | 10 and 50 m | $CH_4$, T, WS and WD |
| 7. Lindenberg | Germany | 52.16 °N | 14.12 °E | 73 m | 10 and 40 m | $CH_4$, T, WD and WD |
| 8. Monte Cimone | Italy | 44.19 °N | 10.69 °E | 2165 m | 8 m | $CH_4$ |
| 9. Norunda | Sweden | 60.08 °N | 17.47 °E | 46 m | 100 m | $CH_4$ |
| 10. Observatoire pérenne de l'environnement | France | 48.56 °N | 5.50 °E | 390 m | 10 and 50 m | $CH_4$, T, WS and WD |
| 11. Puy de Dôme | France | 45.77 °N | 2.96 °E | 1465 m | 10 m | $CH_4$, T, WS and WD |
| 12. Saclay | France | 48.72 °N | 2.14 °E | 160 m | 10, 15 and 100 m | $CH_4$, T, WS and WD |
| 13. Torfhaus | Germany | 51.80 °N | 10.53 °E | 801 m | 10 m | $CH_4$, T, WS and WD |
| 14. Trainou | France | 47.96 °N | 2.11 °E | 131 m | 50 and 100 m | $CH_4$ |

Note. T: air temperature; WS: wind speed; WD: wind direction. $CH_4$ and T were interpolated to roughly 10, 50 and 100 m above ground level, while the simulated WS and WD were calculated based on the model parameters U10 and V10.

### 2.5 Evaluation metrics

There are a number of statistical parameters that can be used to evaluate the performance of atmospheric models, including the correlation coefficient (r), mean bias error (MBE) and root-mean-square error (RMSE). r is a measure of the strength and direction of the linear relationship between simulation and observation, MBE measures the mean difference between





simulation and observation, and RMSE is the square root of the mean squared error between simulation and observation. All three are appropriate over multiple time and space scales and can be calculated as follows:

$$r = \frac{\sum \left[ \left( P_j - \bar{P} \right) x \left( O_j - \bar{O} \right) \right]}{\sqrt{\sum \left( P_j - \bar{P} \right)^2 x \sum \left( O_j - \bar{O} \right)^2}} \tag{4}$$

$$MBE = \frac{1}{n} \sum \left( P_j - O_j \right) \tag{5}$$

$$RMSE = \sqrt{\frac{1}{n} \sum \left( P_j - O_j \right)^2} \tag{6}$$

Here, $j$ represents the pairing of observations ($O$) and predictions ($P$) by site and time. Overbars signify means over site and/or time. $n$ is the number of pairs of observation-prediction values.

In conjunction with the statistics previously mentioned, graphical methods such as time series, scatter plots and Taylor diagrams (Taylor, 2001) were also included to better understand the model behavior over entire ranges of concentrations and gauge performance more fully. For ease of model-satellite data comparison, the satellite data were initially regridded to the model grid and then both satellite and model data were flattened to a one-dimensional array. Overall, as described in section 4, the model simulations of meteorological parameters and methane concentrations were in good agreement with the remote satellite information and near-surface measurements reported at different ICOS sites across Europe. However, several limitations and uncertainties were identified and will help to improve the model's forecast capability in future implementations.

## 3 Model evaluation

### 3.1 Near-surface CH$_4$ concentration

Given that the distances between the model grid points and ICOS sites can be of several kilometers, it is important to highlight that the model evaluation in this section focuses more on the model's ability to reproduce the broad spatial and temporal variability of CH$_4$ over the modeling domain. As mentioned in section 2.2.2, three sets of eight, six and five ICOS stations, with sampling heights between 8.0–16.8 m, 40–50 m, and 100 m, respectively, were selected for comparison with simulated CH$_4$ concentrations interpolated to roughly 10, 50 and 100 m above ground level. Figures 3, 4 and 5 show the monthly mean spatial distributions of observed and simulated CH$_4$ concentrations for the first, second and third vertical



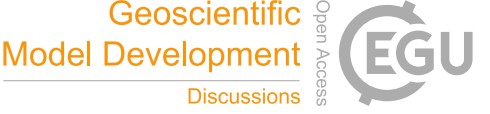

levels, respectively, both data sets averaged over the period from April 1, 2018 to March 31, 2019. Figure S1 in the Supplement shows the monthly mean time series of CH$_4$ concentrations averaged over all ICOS stations and corresponding

model grid points for the three levels.






**Figure 3.** Monthly mean spatial distributions of simulated CH$_4$ concentration interpolated to roughly 10 m (shaded), together with monthly mean CH$_4$ concentrations from the ICOS sites with sampling heights between 8 and 16.8 m (circles), both data sets averaged over the period from April 1, 2018 to March 31, 2019. The concentrations are in ppb and were computed based on quality-controlled ICOS CH$_4$ data for all stations simultaneously.






Overall, $CH_4$ concentrations for the first level were overestimated, mainly during wintertime when model-observation mismatches reached their highest values, between 200 and 300 ppb (see Figure S1 in the Supplement). According to

modeling results in this study, the simulated $CH_4$ concentrations depended largely on the background concentrations, followed by a small contribution from anthropogenic sources. A small month-over-month variation is observed in the $CH_4$ concentrations from ICOS measurements, whereas strong seasonal changes are on the contrary observed in the $CH_4$ concentrations derived from model simulations – seasonal changes in the simulated concentrations are modulated by the anthropogenic sources. No inverse modeling studies of $CH_4$ emissions based entirely on EDGARv6.0 for anthropogenic

sources have been conducted over Europe. However, a recent inversion approach for $CH_4$ emissions over China conducted by Hu et al. (2022) showed that the a posteriori emissions (excluding agricultural soil) decreased by 36% compared to the EDGARv6.0 a priori emission estimates. They also found a 47.1% reduction when it came to $CH_4$ emissions from waste alone. Waste emissions in EDGARv6.0 for 2018 do not have a significant daily and weekly patterns over the year, although emission peaks can be observed in February. Under real conditions, however, the production of $CH_4$ from waste sources

depends not only on the amount of degradable organic matter but also on seasonal weather conditions (Kissas et al., 2022). Uncertainties in EDGAR emissions from other key sectors such as agriculture and energy can also contribute significantly to the overall model-observation discrepancies. For EU27+UK (the 27 European countries and the UK), Solazzo et al. (2021) reported that while $CH_4$ has the best level of accuracy among the three EDGAR greenhouse gases, with only a roughly 10% uncertainty share, the structural uncertainties of the three key sectors in terms of $CH_4$ (agriculture, waste and energy) account

for nearly 90%.

    Comparatively, model-observation discrepancies on $CH_4$ concentration at upper levels (50 and 100 m) were noticeably reduced with increasing height (see Figure S1 in the Supplement). The bias reductions in this case are attributed to a diminishing influence of surface emissions on both magnitude and variability of $CH_4$ concentrations. The top-left panel in Figure S3 in the Supplement shows the reductions in variability as a function of standard deviation, based on a site-

specific comparison. The higher the sampling height (or vertical level), the smaller the model-observation discrepancies in terms of standard deviations are. Despite improvements in terms of variability, correlation coefficients remained quite similar between the three levels, ranging from 0.2 to 0.4 in most cases. Model evaluation of the global CAMS chemical modeling system against ICOS measurements, for the sites here selected and for a period two and a half years from now (https://global-evaluation.atmosphere.copernicus.eu/ch4/ghg/insitu-icos), shows structural correlation coefficients similar to

those found here with WRF-GHG. However, unlike the large positive bias found in this work for the sampling height of 10 m, CAMS does underpredict the observations with model-observation discrepancies ranging from -100 to -200 ppb most of the time. In addition, no bias reduction with increasing height can be noticed in this $CH_4$ product. Input emissions from anthropogenic sources in CAMS simulations are built based on various existing data sets, including nationally reported emissions as well as global estimates (e.g., EDGAR, ECLIPSE and CEDS). As pointed out by Solazzo et al. (2021), the fact





that EDGAR has adopted the IPCC recommendations assures consistency in time and comparability across countries, but conversely, it can facilitate the propagation of uncertainties when similar emission sources are incorporated.

**Figure 4.** Monthly mean spatial distributions of simulated CH$_4$ concentration interpolated to roughly 50 m (shaded), together with monthly mean CH$_4$ concentrations from the ICOS sites with sampling heights between 40 and 50 m (circles), both data

450 sets averaged over the period from April 1, 2018 to March 31, 2019. The concentrations are in ppb and were computed based on quality-controlled ICOS CH$_4$ data for all stations simultaneously.



455

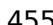

**Figure 5.** Monthly mean spatial distributions of simulated CH$_4$ concentration interpolated to roughly 100 m (shaded), together with monthly mean CH$_4$ concentrations from the ICOS sites with sampling heights of 100 m (circles), both data sets averaged over the period from April 1, 2018 to March 31, 2019. The concentrations are in ppb and were computed based on quality-controlled ICOS CH$_4$ data for all stations simultaneously.





Besides errors in the $CH_4$ emission estimates, inaccuracies in background concentrations and meteorological conditions may have also contributed partly to model-observation discrepancies. With regard to the contribution from background concentrations, boundary conditions in the lowest model layer in CAM-chem are set to the fields specified for Climate

Model Intercomparison Project – Phase 6 (CMIP6) historical conditions and future scenarios provided by Meinshausen et al. (2017). These prescribed $CH_4$ concentrations are then used in the model to overwrite, at each time step, the corresponding model mixing ratios (Lamarque et al., 2012). Thus, the combined effect of using uniform and projected $CH_4$ concentrations as lower boundary conditions in WRF-GHG simulations represents a source of uncertainty and contributes to the model-observation discrepancies. Regarding the meteorological conditions, an unprecedented warmer than normal weather

conditions were observed throughout the study period (Hari et al., 2020), mainly during the 2018-2019 winter season. In fact, model simulations for the period from December 21, 2018 to January 14, 2019 were no included in the model evaluation due to persistent instabilities in vertical winds over central Europe, where a sequence of heavy snowfall events have been observed (e.g., Yessimbet et al., 2022). As can be seen in Figure S2 in the Supplement, the model overpredicted the temperature at 10 m all over the winter, with overpredictions for December (averaged over December 1-20, 2018) and

January (overaged over January 15-31, 2019) being much larger compared to the other winter months. Wind shifts were fairly well represented by the model, but at the same time, it did overpredict wind speed. A site-specific model evaluation in terms of correlation coefficient and standard deviation is provided in Figure S3 in the Supplement.

### 3.2 XCH₄ concentration

Figure 6 shows the temporal mean spatial distributions of $XCH_4$ concentration from SRON RemoTeC-S5P estimates and

WRF-GHG simulations, along with their relative differences, averaged over the period from May 1 to August 31, 2018. Temporal mean spatial distributions by month are shown in Figures S4 to S15 in the Supplement. Differences between simulated $XCH_4$ concentrations with and without smoothing are noticeable. While relative differences between simulated concentrations without smoothing and observational data usually range from -1 to 1% (panel (g) in Figure 6), those between smoothed concentrations and observational data usually range from 1 to 2% (panel (c) in Figure 6). Model-observation

discrepancies in the latter case reached their minimum values during the summer peak season (Figures S6 to S8 in the Supplement), but reached otherwise their maximum values during winter months (Figures S13 to S15 in the Supplement). Model performance for different seasons can be also observed in Figure 7 which shows the monthly variability of observed and simulated $XCH_4$ concentrations over the study region. The lower differences between the satellite measurements and model results without smoothing were related to a $CH_4$ offset (against the anthropogenic emissions contribution), as the

atmospheric layer above the model top (1 hPa) was not vertically integrated in Equation (3). Simulated $CH_4$ concentrations and atmospheric pressures in this case did not experience any smoothing before vertical integration. Regarding the smoothed profiles, despite it was verified that the averaging kernels from satellite retrievals fluctuate slightly up and down around 1 in the troposphere (where much of atmospheric $CH_4$ resides), the smoothing effects in upper levels usually lead to a $XCH_4$ reduction. This reduction often happens because the a priori profile (second term on the right-hand side of Equation (1)) does



not influence the retrieval accuracy significantly (Hu et al., 2016). Since there is no $CH_4$ compensation in this case, then the bigger differences in the $XCH_4$ concentrations can be attributed mainly to an overestimation of anthropogenic emissions, although a systematic bias related to background signals should be also considered. At urban scale, analysis of downwind and upwind concentrations such as the differential column methodology devised by Chen et al. (2016) can be applied for minimizing the influence of background signals (e.g., Zhao et al., 2022); however, its application at a continental scale would

require a high-resolution modeling configuration as well as a dense network of spectrometers. Data gaps such as those observed in central and southern Europe (see panels (a) or (e) in Figure 6) are often produced as a consequence of applying regridding techniques to sparse data sets.

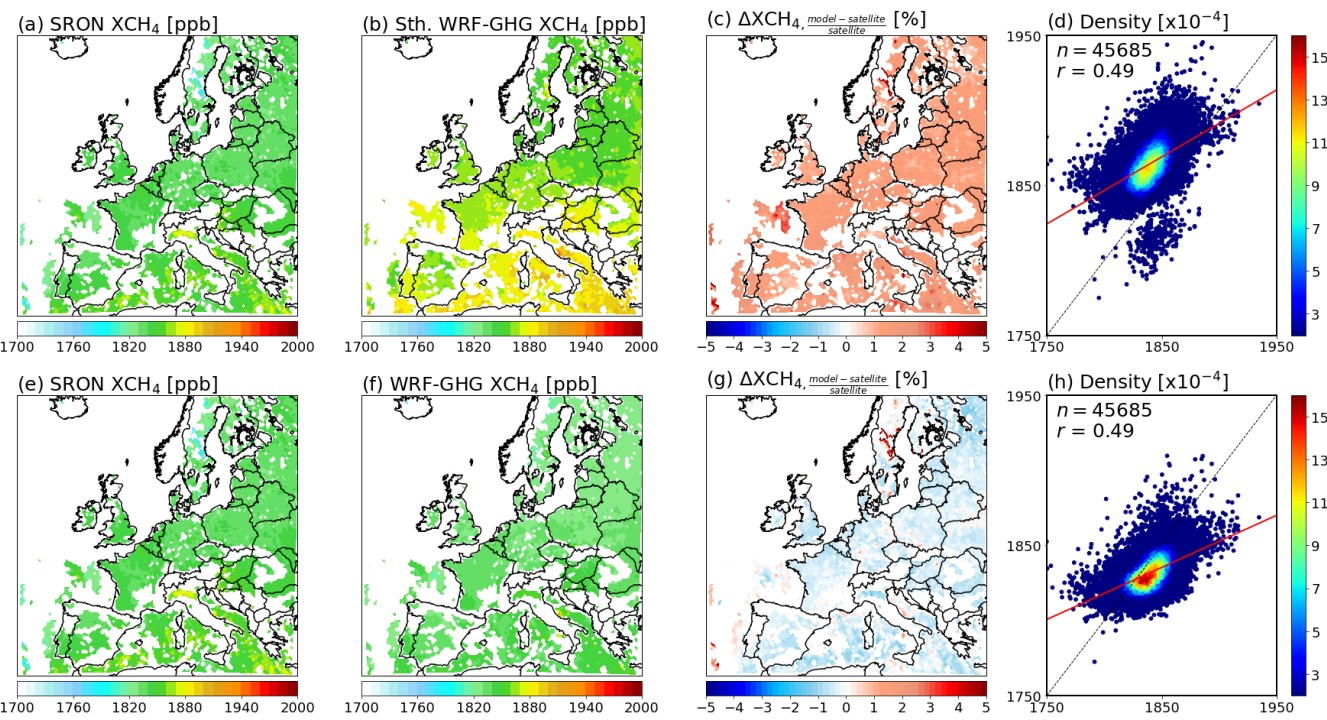

**Figure 6.** Temporal mean spatial distributions of $XCH_4$ concentration from SRON RemoTeC-S5P (panels (a) and (e)) and WRF-GHG estimates with and without smoothing (panels (b) and (f), respectively), along with their relative differences (panels (c) and (g)), averaged over the period from May 1 to August 31, 2018. The simulated mean $XCH_4$ concentrations are calculated on the basis of the closest model times to the S5P crossing times. The panels on the right show the scatterplots of

observed and simulated $XCH_4$ concentrations, together with the number of pairs of observation-model values and domain-wide correlation coefficient.



Zhao et al. (2019) applied the WRF-GHG model to analyze XCH$_4$ observations over Berlin for a period during summertime, and found a bias in the simulated XCH$_4$ concentration of around 2.7%. In that case, they updated the boundary conditions

with information from CAMS instead of CAM-chem, used EDGARv4.1 emission estimates for anthropogenic sources and compared the model results against total column measurements from a network of five spectrometers. Based on the smoothed concentrations in this work, relative differences between 1 and 2% were often found for summer peak season, when the anthropogenic sources had their minimum contributions to the XCH$_4$ concentration. For winter months, the differences were found to range from 2 to 3%, similar to those found by Zhao et al. (2019) for summertime. Despite a model

overestimation of near-surface CH$_4$ concentrations on the order of 200-300 ppb is observed during wintertime, the model-observation discrepancies on XCH$_4$ concentration ranged roughly from 40 to 60 ppb. The higher model-observation discrepancies during winter months suggest that a more refined inverse analysis assessment will be required for this season. A recent joint inversion of CH$_4$ and $\delta^{13}$C-CH$_4$ conducted by Basu et al. (2022) for periods of relatively stability (2000-2006) and growth (2008-2014) in atmospheric CH$_4$ suggests a significant reduction in the a priori CH$_4$ emission estimates from

fossil and microbias sources over northern extra-tropic regions. Bias in simulated XCH$_4$ concentrations over water bodies, namely the Mediterranean Sea, Bay of Biscay and small portions of the Atlantic Ocean adjacent to Spain and Portugal, is of similar magnitude as that found over land. Thanks to the TROPOMI's wide swath, the SRON S5P-RemoTeC XCH$_4$ product v17 provides new opportunities to look into sensitivity of CH$_4$ signals to surface emissions in the Mediterranean Sea (e.g., CH$_4$ emissions from oil and gas platforms).


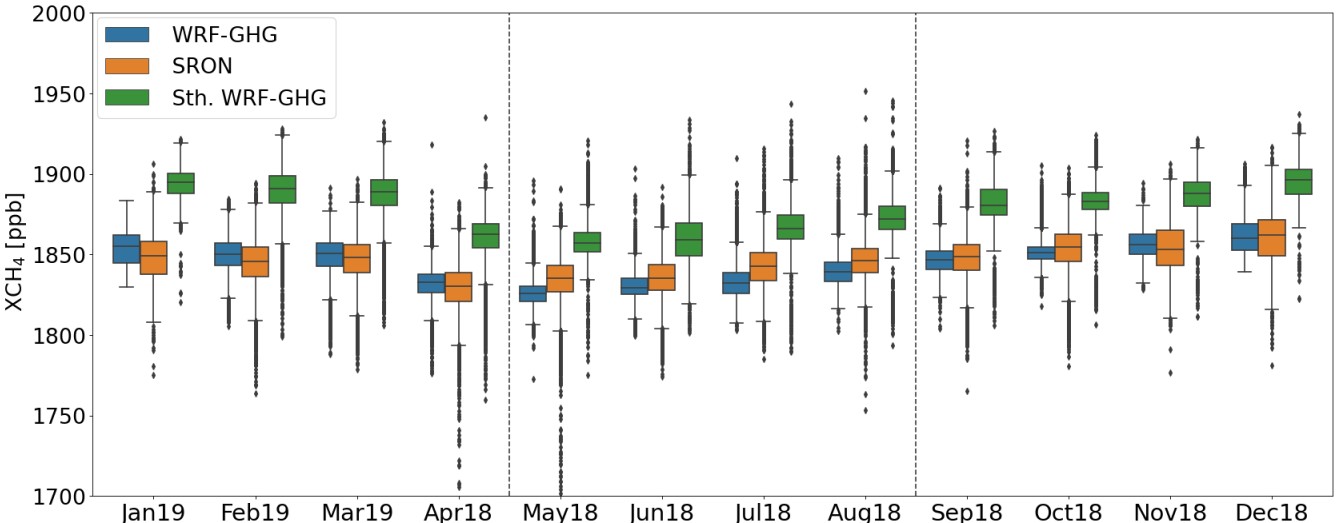

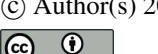



**Figure 7.** Monthly boxplots of observed and simulated XCH$_4$ concentrations with and without smoothing for the period from April 01, 2018 to March 31, 2019. The months from May to August of 2018 (between dashed lines) were selected for model evaluation of the contribution of anthropogenic sources to the XCH$_4$ concentration, discussed ahead in section 4.

With regard to temporal variability, a clear annual cycle of the XCH$_4$ interquartile range can be noticed regardless of its 565 smooth month-over-month variation along the year (orange boxes in Figure 7). Both sets of simulated XCH$_4$ concentrations, i.e. the simulated profiles with and without smoothing, represented fairly well this cycle although with a less dispersion (length of the box). The simulated XCH$_4$ concentrations without smoothing show even a less-dispersed interquartile range (blue boxes in Fig. 7) compared to that of the smoothed concentrations (green boxes in Figure 7). In addition, the minimum concentrations in the simulated interquartile ranges are delayed (May) compared to observations (April), with the same 570 happening in terms of medians. Based on the smoothed concentrations, model-observation discrepancies reached their maximum values during winter months (differences in median concentrations between 40-50 ppb), while they reached their minimum values during summer peak season (differences in median concentrations between 20-30 ppb). As discussed in section 3.2, the better XCH$_4$ representation with the simulated profiles without smoothing responded to a CH$_4$ compensation, as the atmospheric layer above the model top (1 hPa) was not vertically integrated in Equation (3). Since there is no a CH$_4$ 575 compensation with the smoothed profiles, then the bigger differences in the XCH$_4$ concentrations can be attributed mainly to an overestimation of anthropogenic emissions. A systematic bias related to the background concentrations, however, should be also embedded in the model bias in both cases. Modeling studies using CAMS suggest that an offset between model concentrations and observations needs to be taken into account previously in the boundary conditions (Zhao et al., 2019; Galkowsky et al., 2021). Looking at the other 50% of data, including outliers, it can be also observed a similar behavior with 580 observations spread out further than simulated concentrations. A number of outliers with concentrations below 1750 ppb have been observed in April and May of 2018, although the reason why they occur in the months with the lowest CH$_4$ concentrations needs to be further investigated. Statistical metrics of the model-observation comparison indicate, overall, a better model performance for summer months, with correlation coefficients and root-mean-square errors ranging from 0.4-0.5 and 27-30 ppb, respectively (see Table 4).

585

590



**Table 4.** Overall WRF-GHG performance against space-based $XCH_4$ observations.

| Month | WRF-GHG/SRON S5P | | | | Sth. WRF-GHG/SRON S5P | | | |
|---|---|---|---|---|---|---|---|---|
| | n | RMSE | MBE | r | n | RMSE | MBE | r |
| January 2019 | 869 | 17.32 | 6.86 | 0.43 | 869 | 50.42 | 46.36 | 0.15 |
| February 2019 | 9893 | 16.05 | 5.34 | 0.35 | 9893 | 48.44 | 45.09 | 0.29 |
| March 2019 | 8691 | 15.95 | 1.80 | 0.31 | 8691 | 44.22 | 40.23 | 0.24 |
| April 2018 | 15430 | 16.35 | 2.77 | 0.32 | 15430 | 35.88 | 31.76 | 0.39 |
| May 2018 | 16817 | 17.38 | -8.28 | 0.25 | 16817 | 28.63 | 24.13 | 0.31 |
| June 2018 | 8590 | 12.08 | -4.76 | 0.54 | 8590 | 27.77 | 24.39 | 0.53 |
| July 2018 | 9546 | 15.49 | -9.47 | 0.52 | 9546 | 28.84 | 24.93 | 0.44 |
| August 2018 | 10732 | 13.81 | -6.77 | 0.47 | 10732 | 29.96 | 26.64 | 0.40 |
| September 2018 | 10114 | 12.02 | -0.98 | 0.44 | 10114 | 36.97 | 34.27 | 0.36 |
| October 2018 | 10707 | 13.21 | -2.09 | 0.34 | 10707 | 33.40 | 29.72 | 0.20 |
| November 2018 | 2917 | 14.93 | 3.01 | 0.46 | 2917 | 37.95 | 33.41 | 0.23 |
| December 2018 | 951 | 16.90 | 1.25 | 0.51 | 951 | 39.97 | 34.86 | 0.37 |

Note. n: number of pairs of observation-model values; RMSE: root-mean-square error (in ppb); MBE: mean bias error (in ppb); r: correlation coefficient.

## 4 $XCH_4$ concentration from anthropogenic sources

Contribution of anthropogenic emissions to the $XCH_4$ concentration is calculated based on the months with the best model performance, between May and August of 2018 (see Figure 4). Figure 8 shows the temporal mean spatial distributions with and without smoothing of simulated $XCH_4$ concentrations, $XCH_4$ enhancement above background (EAB) concentrations, $XCH_4$ enhancement from human activities (EHA) concentrations, and contribution of anthropogenic sources to the $XCH_4$ concentration. Model results suggest that $XCH_4$ EHA concentrations as high (or even higher) as those found over high $CH_4$ emitting countries in western Europe can accumulate over countries in central and southern parts of Europe during summer months (see panels (c) and (g) in Figure 8). The $XCH_4$ EAB concentrations (panels (b) and (f)) depended almost entirely on the $CH_4$ contribution from human activities (panels (c) and (g)), result that is in line with previous studies conducted over urban areas in central Europe (e.g., Zhao et al., 2019; Zhao et al., 2022). However, the anthropogenic sources contribute only up to 2% to the $XCH_4$ concentration (panels (d) and (h)), with most part of $XCH_4$ coming from background signals.

$XCH_4$ signals from natural sources (wetlands and termites) and biomass burning were not relevant during the study period. According to Kaplan (2002), potential natural wetlands in the 30 km modeling domain concentrate over the Baltic countries, Belarus and western regions of Russia. Among the factors that could have negatively influenced the accumulation of biospheric $CH_4$ in the atmosphere over the study region are: a less $CH_4$ formation tied to the extremely dry season in summer 2018 over central and northern Europe (Rousi et al., 2022); a $CH_4$ compensation by soil uptake processes; and



transport mechanisms. Yu et al. (2022) suggest that northern temperate wetland emissions in Russia show strong sensitivity to both hydrology and temperature. On the other hand, winds may disperse $CH_4$ concentrations out of the study region, thus

reducing drastically the $XCH_4$ concentrations over specific regions. Both the observed and simulated wind patterns over central Europe show that, between May and August of 2018, air masses flowed mostly southeast-southwest (see Figure S2 in the Supplement), deflecting that way most of the air coming from wetland areas.

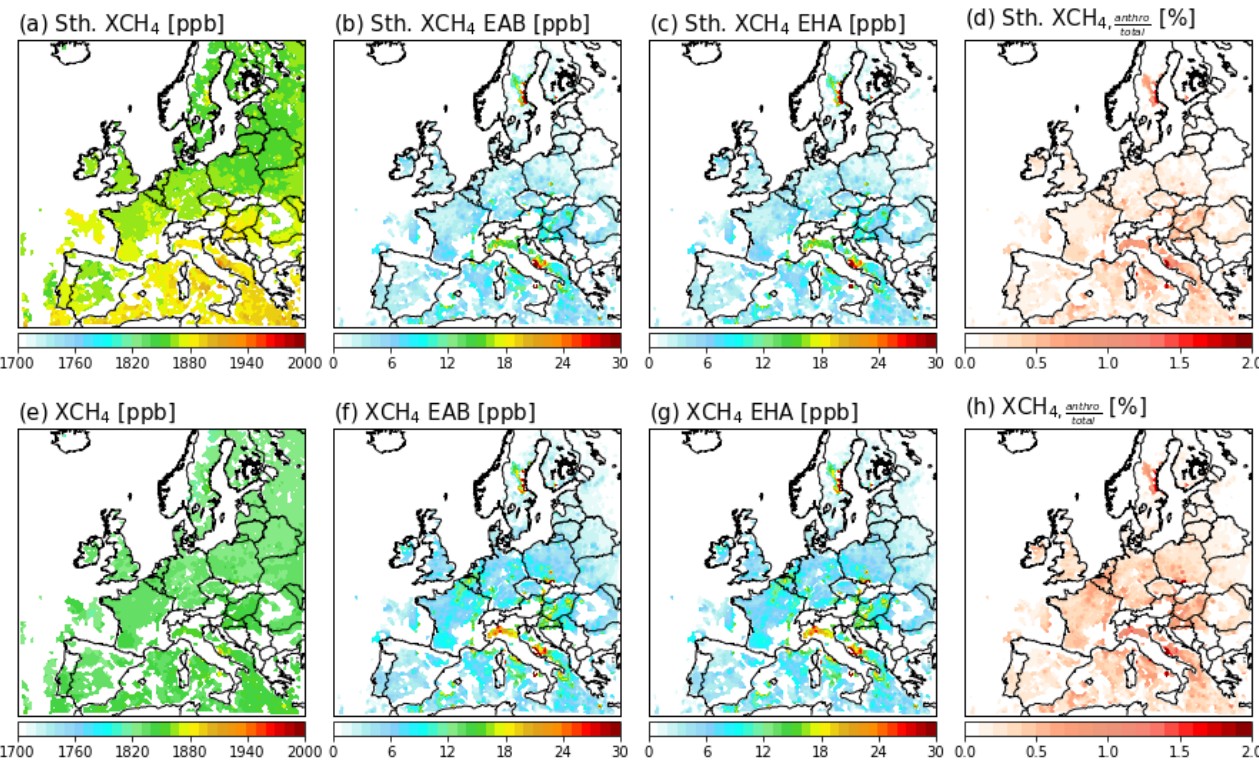

**Figure 8.** Temporal mean spatial distributions with and without smoothing of simulated $XCH_4$ concentration (panels (a) and (e)), $XCH_4$ enhancement above background (EAB) (panels (b) and (f)), $XCH_4$ enhancement from human activities (EHA) (panels (c) and (g)), and contribution of anthropogenic sources to the $XCH_4$ concentration (panels (d) and (h)). Concentrations were averaged for grid points with satellite measurements during the period from May 1 to August 31, 2018, months with the best model performance for smoothed concentrations (see Fig. 4).

**5 Summary and conclusions**

A new $CH_4$ inversion system for Europe is being implemented in order to evaluate $CH_4$ emission estimates from different sources, with a focus on anthropogenic activities. In this first part, the forward modeling component of the system is





introduced and evaluated against $CH_4$ column-averaged dry air mole fractions ($XCH_4$) and near-surface $CH_4$ observations. To that end, sets of 97-hour simulations for one-year simulation period from April 01, 2018 to March 31, 2019, were run

using the WRF greenhouse gases model coupled to a multipurpose global database of $CH_4$ anthropogenic emissions. $CH_4$ fluxes from biogenic sources were calculated online in the simulations, whereas fluxes from biomass burning were externally prepared based on a satellite-based emissions preprocessor. Model results were evaluated against Netherlands Institute for Space Research (SRON) S5P-RemoTeC $XCH_4$ (v17) concentrations, as well as against $CH_4$ Level 2 data from Integrated Carbon Observation System (ICOS) stations. Simulated $XCH_4$ concentrations without taking into account the a

priori information and averaging kernels from satellite retrievals were also computed to evaluate smoothing effects.

Model-observation discrepancies on near-surface $CH_4$ concentration (10 m) indicate a significant overestimation on the order of 200-300 ppb during winter months. Comparatively, model-observation discrepancies on $CH_4$ concentration at upper levels (50 and 100 m) were noticeably reduced with increasing height. The bias reductions in this case are attributed to a diminishing influence of surface emissions on both magnitude and variability of $CH_4$ concentrations. In terms of $XCH_4$, a

better representation was found with the simulated profiles without smoothing – it was related to a $CH_4$ offset (against the anthropogenic emissions contribution) as the atmospheric layer above the model top was not vertically integrated in Equation (3). Based on the smoothed concentrations, model-observation discrepancies reached their maximum values during winter months (differences in median concentrations between 40-50 ppb), while they reached their minimum values during summer peak season (differences in median concentrations between 20-30 ppb). Domain-wide correlation coefficients and

root-mean-square-errors ranged from 0.4 to 0.5 and from 27 to 30 ppb, respectively, for summer months, and from 0.1 to 0.4 and from 33 to 50 ppb, respectively, for winter months. The higher model-observation discrepancies on $XCH_4$ concentration found during winter months are largely related to a significant overestimation of anthropogenic emissions; however, a systematic bias related to background signals should be also embedded in the model bias, in both scenarios with and without smoothing. The $XCH_4$ enhancement above background concentrations depended almost entirely on the $CH_4$ contribution

from anthropogenic sources; however, these sources contributed with only up to 2% to the $XCH_4$ concentration. $XCH_4$ signals from natural sources (wetlands and termites) and biomass burning were not relevant during the study period.

The results found in this study are in line with previous studies conducted over urban areas in central Europe, and thus, demonstrate a huge and under explored potential for $CH_4$ inverse modeling using updated TROPOMI $XCH_4$ data sets in large-scale applications. As discussed in section 3, model results suggest a significant overestimation of anthropogenic

emissions during winter months. Then, for a better constraint of monthly country-scale fluxes of $CH_4$, an inverse analysis method taking full advantage of all satellite data available for a given month might provide much more accurate emission estimates. Ongoing work is being conducted in this direction and will be published in a second part.

**Code and data availability**

The WRF-Chem model code version 4.3 is freely distributed by NCAR at

https://www2.mmm.ucar.edu/wrf/users/download/get_source.html. The WRF-Chem preprocessor tools anthro_emis,





fire_emis and mozbc are provided by NCAR at https://www2.acom.ucar.edu/wrf-chem/wrf-chem-tools-community. Run control files, preprocessing and postprocessing scripts to replicate the modeling results in this work, i.e., the scripts used to prepare the emission files, run the model, and calculate the total column concentrations, are all freely accessible at https://github.com/alvv1986/AUMIAv1.0.

## Author contributions

AVV developed the research design and methodology, performed the simulations, analysis, and wrote the manuscript. CK contributed with fruitful discussions on the satellite data and model evaluation, and also leads the project that produced this research work. NRB contributed with satellite data processing. JN contributed with enriched suggestions across different stages of the study. All authors provided critical feedbacks and helped shape the research, analysis and manuscript.

## Competing interests

The authors declare that they have no conflict of interest.

## Acknowledgements

This research has been supported by the Villum Fonden (Grant No. 40709). AVV and CK thank the LUMI supercomputer access grants received through the Danish e-Infrastructure Cooperation/DeiC-AU-N5-000026 (Validation of the Danish methane emission). The authors acknowledge the free availability of the WRF-Chem model, in-situ data from the ICOS network, $XCH_4$ observations from the Netherlands Institute for Space Research (SRON) S5P-RemoTeC $XCH_4$ product version 17, and ERA5 fields in the Copernicus Climate Data Store. We acknowledge use of the WRF-Chem preprocessor tools and data sets provided by the Atmospheric Chemistry Observations & Modeling Lab (ACOM) of NCAR. We additionally thank Mario Gavidia-Calderon for sharing his python tools (https://github.com/quishqa), some of which have been customized for use in this work.

## Appendix A: Model running process

As a detailed description on how to run WRF-GHG can be found in Beck et al. (2011), only the initialization process, which can vary depending on specific requirements, is summarized here. Firstly, moving simulations of 97 hours were performed automatically for each month so that the number of 97-hour simulations in a given month constitute a cycle in our automated bash routines. Each cycle begins, through its first moving simulation, with initial and boundary conditions previously prepared from a 7-day simulation which ends at the initialization time of the cycle, at YYYY-MM-DD 00:00:00 (according to WRF-GHG date and time format). Most of this 7-day simulation is discarded as spin-up time and only the last hour is saved to be used as initial conditions in the first moving simulation. Then, when the first moving simulation ends, the second one begins right after it with initial conditions prepared from the first moving simulation at YYYY-MM-D+1 00:00:00, and



goes ahead up to complete a 97-hour simulation length at YYYY-MM-D+5 00:00:00. The third moving simulation will begin right after the second one ends, with initial conditions prepared from the second moving simulation at YYYY-MM-D+2 00:00:00, and will go ahead up to complete a 97-hour simulation length at YYYY-MM-D+6 00:00:00. This process will continue up to complete the cycle, with the same procedures being applied to the other 11 remaining cycles. The boundary conditions are prepared from CAM-chem data during the preprocessing part in each moving simulation. With this

methodology, all satellite data available for a given month could potentially be ingested in sets of up to 73-hour backward in time simulations. As the first day of each moving simulation is used as spin-up time, it is discarded and only the second day is used for model evaluation. The simulations were executed on LUMI (Large Unified Modern Infrastructure), which is a pan-European pre-exascale supercomputer able to provide computing power of up to 552 petaflops.

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
