# Peer review of "Implementation of a Satellite-Based Tool for the Quantification of CH4 emissions over Europe (AUMIA v1.0) – Part 1: Forward Modeling Evaluation against Near-Surface and Satellite Data"

_Geoscientific Model Development, 2023_

## Author Response (AR2)

**Response to reviewers**

Ref.: GMD-2023-9 | Model evaluation paper

**Inverse Modeling of CH$_4$ emissions over Europe, Part I: Forward Modeling Evaluation against Near-Surface and Satellite Data**

Dear Topical Editor,

First of all, we want to thank the two anonymous reviewers for their insightful comments and suggestions which have been addressed accordingly. We hope now that our manuscript does meet the requirements of the journal, and thus can be accepted for final publication in GMD. Please find below a detailed, point-by-point response (in **bold**) to all reviewer's comments (in *italics*). Original sentences/paragraphs and the changes are indicated in blue as follows: **"original" ➔ "modified"**

**Referee #1**

*General comments:*

*My main concern with this work is that it intends to introduce a new inversion framework for TROPOMI CH4 data, itself somewhat incremental, but the authors have decided to split the paper in to 2 parts. With this first part only concerning the forward model, it is difficult to assess and explain the differences compared to observations. The forward model fails to capture the variability observed by the ground-based ICOS network and also fails to match the satellite total column observations. The issues may be with the prior as the authors suggest (and which would be demonstrated by actually showing the inversion results) but*

*when the authors apply the averaging kernels (as should be done), this "smoothing" effect leads to much poorer comparisons and there's not a sufficient explanation for this.*

*Unfortunately I believe that the attempt to split the paper into two has led to this first part being particularly weak and lacking. There's clearly significant work that has gone in to this study but I would recommend that the authors consider publishing it as a whole, therefore being able to back up their speculation with quantitative inversion results.*

**We agree that it is difficult to assess and explain the model-observation differences, and also that the model fails to capture the ground-based ICOS and TROPOMI observations. This is basically why inverse modeling systems need to be either developed or constantly improved. Even comprehensive global forecasting systems such as the CAMS system ([https://global-evaluation.atmosphere.copernicus.eu/ch4/ghg/insitu-icos](https://global-evaluation.atmosphere.copernicus.eu/ch4/ghg/insitu-icos)) do predict methane discrepancies on the order of 100-300 ppb for the same ICOS stations we used in the model evaluation (see a brief discussion on this in the manuscript lines 412-417). In addition, the domain-wide correlation coefficients (0.4–0.5) and root-mean-square errors (27–30 ppb) we obtained for $XCH_4$ (taking into account the smoothing effect) are in line with previous studies conducted over specific locations in Central Europe (e.g. Zhao et al., 2019, 2022; Tsuruta et al., 2023).**

**The fact that the smoothing effect leads to a much poorer comparisons has also been discussed on the manuscript lines 513-522. The lower differences between the satellite estimates and the simulated concentrations without smoothing are related to an artificial methane offset. This methane offset is a result of not integrating the atmospheric layer above the model top (~1hPa), thus lowering the total-column concentrations artificially. Compared to the case without smoothing, the simulated concentrations with smoothing take into account the atmospheric layer above the model top and thus higher total-column concentrations are expected in this case. It is worth**

reminding that in order to compare methane profiles from atmospheric models against satellite estimates, the model profiles need to be previously smoothed using the a priori information and averaging kernels from the satellite retrievals. The smoothing effect is not usually included as part of model evaluation in methane modeling papers; however, we do believe that including a discussion on this effect is relevant to the modeling community.

With all of the state-of-the-art modeling tools and improved TROPOMI observations that have been applied for evaluating methane concentrations over Europe, but, most importantly, the consistent results (together with a better understanding of how to handle TROPOMI observations) achieved at this first stage, we disagree that our manuscript is weak and lacking. We understand the reviewer's point of view that publishing the modeling system as a whole would be better, especially considering that the inversion part by itself captures much more attention than forward simulations; however, we do think that gradual dissemination of findings and contributions can establish a foundation for subsequent parts. Submitting a forward modeling evaluation paper first will allow us to receive timely feedback and suggestions that can inform any necessary improvements or modifications that need to be made before delving into further aspects of the backward modeling part (e.g. by identifying the problem with not including the top layer in the integration, which the inversion part would likely just have solved by adjusting the state vector).

Tsuruta, A. et al.: $CH_4$ Fluxes Derived from Assimilation of TROPOMI $XCH_4$ in CarbonTracker Europe-$CH_4$: Evaluation of Seasonality and Spatial Distribution in the Northern High Latitudes, Remote Sensing, 15, 1620, doi:10.3390/rs15061620, 2023.

**Zhao, X. et al.: Understanding greenhouse gas (GHG) column concentrations in Munich using WRF, Atmos. Chem. Phys., doi:10.5194/acp-2022-281, 2022.**

**Zhao, X. et al.: Analysis of total column $CO_2$ and $CH_4$ measurements in Berlin with WRF-GHG, Atmos. Chem. Phys., 19, 11279-11302, 2019.**

*Specific comments:*

*Abstract: The authors appear to very strongly oversell their work with the statement that "The results found in this study contribute with a new model evaluation of methane concentrations over Europe, and demonstrate a huge and under explored potential for methane inverse modeling using improved TROPOMI products in large-scale applications." Inverse modelling of methane is a very active area with a strong track record from a number of European groups. Many groups have published inversion results using TROPOMI data and indeed, there are large European projects in this area.*

**We agree that inverse modeling of methane is a very active area and that there are large European projects in this area. In line with those projects, the second part of this work will represent a first effort in years to update the methane emission estimates for Denmark. The first part of the sentence "The results found in this study contribute with a new model evaluation of methane concentrations over Europe" is true in the sense that new EDGAR emission estimates (Ferrario et al., 2021) together with improved TROPOMI observations (Lorente et al., 2022) were used to evaluate the methane concentrations over Europe. Regarding the second part "and demonstrate a huge and under explored potential for methane inverse modeling using improved TROPOMI products in large-scale applications", it can be supported by the domain-wide statistical metrics which are in line with previous studies conducted over specific locations in Central Europe (e.g. Zhao et al., 2019, 2022; Tsuruta et al., 2023). The more satellite**

observations are improved, the more accurate the inversion estimates tend to be. The term "and under explored" has been removed to not insinuate the TROPOMI data we used here have been available for so long and that even so it's been under explored.

Lines 34-36 in the new version of the manuscript: "The results found in this study contribute with a new model evaluation of methane concentrations over Europe, and demonstrate a huge and under explored potential for methane inverse modeling using improved TROPOMI products in large-scale applications." ➔ "The results found in this study contribute with a new model evaluation of methane concentrations over Europe, and demonstrate a huge potential for methane inverse modeling using improved TROPOMI products in large-scale applications."

Lines 652-654 in the new version of the manuscript: "The results found in this study are in line with previous studies conducted over urban areas in central Europe, and thus, demonstrate a huge and under explored potential for $CH_4$ inverse modeling using updated TROPOMI $XCH_4$ data sets in large-scale applications." ➔ "The results found in this study are in line with previous studies conducted over urban areas in central Europe, and thus, demonstrate a huge potential for $CH_4$ inverse modeling using updated TROPOMI $XCH_4$ data sets in large-scale applications."

Also, the proper EDGARv6.0 reference is now included in the manuscript, lines 174-175 in the new version of the manuscript: "Anthropogenic fluxes of $CH_4$ (not including biomass burning sources) are externally prepared based on the Emissions Database for Global Atmospheric Research (EDGAR) version 6 Greenhouse Gas Emissions (Crippa et al., 2021). " ➔ "Anthropogenic fluxes of $CH_4$ (not including biomass burning sources) are externally prepared based on the Emissions Database for Global Atmospheric Research (EDGAR) version 6 Greenhouse Gas Emissions (Ferrario et al., 2021). "

Ferrario, M. et al.: EDGAR v6.0 Greenhouse Gas Emissions. European Commision, Joint Research Centre (JCR) [Dataset] PID: http://data.europa.eu/89h/97a67d67-c62e-4826-b873-9d972c4f670b, 2021.

Lorente, A. et al.: Evaluation of the methane full-physics retrieval applied to TROPOMI ocean sun glint measurements, Atmos. Meas. Tech., 15, 6585-6603, 2022.

Tsuruta, A. et al.: $CH_4$ Fluxes Derived from Assimilation of TROPOMI $XCH_4$ in CarbonTracker Europe-$CH_4$: Evaluation of Seasonality and Spatial Distribution in the Northern High Latitudes, Remote Sensing, 15, 1620, doi:10.3390/rs15061620, 2023.

Zhao, X. et al.: Understanding greenhouse gas (GHG) column concentrations in Munich using WRF, Atmos. Chem. Phys., doi:10.5194/acp-2022-281, 2022.

Zhao, X. et al.: Analysis of total column $CO_2$ and $CH_4$ measurements in Berlin with WRF-GHG, Atmos. Chem. Phys., 19, 11279-11302, 2019.

*L65 – This ignores some of the TIR instruments that measure CH4, IASI being maybe the most relevant here. Some mention of these should be made and then an explanation on why the focus is on the SWIR instruments.*

**The second paragraph in the introduction section describes what satellite platforms have been mostly used for evaluating methane simulations, and also mentions the platforms that are still in operation. In the case of IASI, it has been mostly used for evaluating carbon monoxide and ozone simulations, with just a few studies on methane having been reported.**

**Anyway, we agree that it is worth including IASI in the introduction section, lines 57-62 in the new version of the manuscript: "Such comparative studies have**

focused mostly on CH$_4$ column-averaged dry air mole fractions (hereafter referred to as XCH$_4$ concentrations) from the SCanning Imaging Absorption spectroMeter for Atmospheric ChartographY (SCIAMACHY) and Thermal And Near-infrared Sensor for carbon Observation (TANSO) instruments onboard the Environmental Satellite (EnviSat) and Greenhouse gases Observing SATellite (GOSAT), respectively." ➔ "Such comparative studies have focused mostly on CH$_4$ column-averaged dry air mole fractions (hereafter referred to as XCH$_4$ concentrations) from the SCanning Imaging Absorption spectroMeter for Atmospheric ChartographY (SCIAMACHY), Thermal And Near-infrared Sensor for carbon Observation (TANSO), and Infrared Atmospheric Sounding Interferometer (IASI) instruments onboard the Environmental Satellite (EnviSat), Greenhouse gases Observing SATellite (GOSAT), and Meteorological Operational (Metop-A -B and -C) satellites, respectively."

**Also in lines 65-67:** "TANSO provides more mature but sparser XCH$_4$ concentrations than TROPOMI, and is together with TROPOMI the only two satellite instruments that remain operational since they were launched in 2009 and 2017, respectively." ➔ "TANSO and IASI provide more mature but sparser XCH$_4$ concentrations than TROPOMI, and are together with TROPOMI the only three satellite instruments that remain operational since they were launched in 2009, 2012 (Metop-B) and 2017, respectively."

**There is no actually a specific reason why we selected a SWIR instrument, but taking advantage of a much more sparser and high-resolution XCH$_4$ product.**

*L74: Tsuruta et al. (2023) would appear to be a very relevant reference, given it involves TROPOMI inversions over Europe, that is omitted. Some discussion of how this work relates/compares to that should be undertaken.*

**This work had not been included in the discussion section because it was not available by the time when we submitted our manuscript to GMD. New discussions on Tsuruta et al. (2023) results have now been included in several parts throughout the manuscript, lines 397-401 in the new version of the manuscript:** "...anthropogenic sources. No inverse modeling studies of $CH_4$ emissions based entirely on EDGARv6.0 for anthropogenic sources have been conducted over Europe. However, a recent inversion approach for $CH_4$ emissions over China…" ➔ "...anthropogenic sources. Using EDGARv6.0 $CH_4$ fluxes as the a priori emission estimates and two sets of TROPOMI-based $XCH_4$ observations, the global inversion approach conducted by Tsuruta et al. (2023) showed that over central Europe the anthropogenic $CH_4$ emissions would be slightly overestimated, mainly during spring and autumn. However, higher emission estimates are otherwise found when ground-based data is used to drive the inversions. The inversion approach for $CH_4$ emissions over China…"

**And lines 598-605:** "$XCH_4$ signals from natural sources (wetlands and termites) and biomass burning were not relevant during the study period. According to Kaplan (2002), potential natural wetlands in the 30 km modeling domain concentrate over the Baltic countries, Belarus and western regions of Russia. Among the factors that could have negatively influenced the accumulation of biospheric $CH_4$ in the atmosphere over the study region are: a less $CH_4$ formation tied to the extremely dry season in summer 2018 over central and northern Europe (Rousi et al., 2022); a $CH_4$ compensation by soil uptake processes; and transport mechanisms. Yu et al. (2022)..." ➔ "$XCH_4$ signals from natural sources (wetlands and termites) and biomass burning were not relevant during the study period. The inversion estimates for 2018 conducted by Tsuruta et al. (2023) showed that, compared to the anthropogenic emissions, the wetland emissions over central Europe were small, mainly during summer months when biogenic fluxes

reached their minimum values. Among the factors that could have negatively influenced the accumulation of biospheric $CH_4$ in the atmosphere over the study region are: a less $CH_4$ formation tied to the extremely dry season in summer 2018 over central and northern Europe (Rousi et al., 2022); a $CH_4$ compensation by soil uptake processes as the fluxes are dominated by mineral soils which are mostly net sink of $CH_4$ (Tsuruta et al., 2023); and transport mechanisms. According to the Kaplan wetland map, potential natural wetlands in Europe concentrate over western regions of Russia. Yu et al. (2022)..."

Kaplan, J. O.: Wetlands at the last Glacial Maximum: Distribution and methane emissions, Geophys. Res. Lett., 29, 1079, 2002.

Rousi, E. et al.: The extremely hot and dry 2018 summer in central and northern Europe from a multi-faceted weather and climate perspective, EGUsphere, doi:10.5194/egusphere-2022-813, 2022.

Tsuruta, A. et al.: $CH_4$ Fluxes Derived from Assimilation of TROPOMI $XCH_4$ in CarbonTracker Europe-$CH_4$: Evaluation of Seasonality and Spatial Distribution in the Northern High Latitudes, Remote Sensing, 15, 1620, doi:10.3390/rs15061620, 2023.

Yu, X. et al.,: A high-resolution satellite-based map of global methane emissions reveals missing wetland, fossil fuel and monsoon sources, EGUsphere, doi:105194/egusphere-2022-948, 2022.

*L98: "carefully selected" – How? Why? What criteria?*

The main criteria for selecting the two-week periods for model sensitivity tests was to have at least 75% of days with TROPOMI data covering large portions of Europe. Large numbers of observation/model pairs (that spread out across the modeling

domain) allow to perform a more representative domain-wide statistical evaluation. The one-year period from April 01, 2018 to March 31, 2019 was selected because of the following reasons: 1) availability of TROPOMI operational data and the improved TROPOMI data from March 2018 onwards; 2) evaluate the most recent EDGARv6.0 emissions for methane (2018); and 3) avoid sustained irregular scenarios in terms of emissions, e.g., fire outbreaks in most part of 2019 and emission reductions associated with COVID-19 lockdowns in 2020 and 2021, both at global scale. The 2018 summer was particularly interesting to focus on because of the larger than average number of cloud-free days over Denmark and most part of Europe.

Section 2.3 Experimental design was rewritten and now it includes the criteria for selecting the study periods for the model sensitivity tests, lines 242-250 in the new version of the manuscript: "Initially, a model sensitivity analysis for evaluation of model parameterizations such as planetary boundary layer and cumulus clouds, as well as global forcings for $CH_4$ concentration, was carried out over several two-week periods in 2018 and 2019. Then, based on the model configuration that best fit the satellite data, one-year simulation period from..." → "Initially, a model sensitivity analysis for evaluating physics schemes such as planetary boundary layer and cumulus clouds, and global forcings for meteorological fields and $CH_4$ concentration, was carried out over several two-week periods in 2018 and 2019. Each of these two-week periods were previously examined to have at least 75% of days with TROPOMI $XCH_4$ data covering large portions of Europe. As a result, the physics schemes Yonsei University (YSU) for planetary boundary layer and Kain-Fritsch for cumulus clouds, together with initial and boundary conditions from the European Centre for Medium-Range Weather Forecasts (ECMWF) Reanalysis v5 (ERA5) model (Hersbach et al., 2020), for meteorological processes, and from the NCAR Community Atmosphere Model with

Chemistry (CAM-chem) (Lamarque et al., 2012; Emmons, et al., 2020), for background concentrations of $CH_4$, were selected and then used to perform a one-year simulation period from…"

And for the one-year simulation period, lines 250-253: "This period was defined based on the following criteria: i) availability of TROPOMI $XCH_4$ data, ii) latest available year data of EDGARv6.0 emissions for $CH_4$, and iii) no occurrence of sustained and irregular scenarios in terms of emissions (e.g., large-scale fire outbreaks and emission reductions associated with COVID-19 lockdowns)."

*L100: I think there needs to be a strong justification for doing this as a 2-part paper and I'm struggling to see why it needed to be split. Can the authors please expand upon the rationale for this.*

As previously mentioned, we understand the reviewer's point of view that publishing the modeling system as a whole would be better, especially considering that the inversion part by itself captures much more attention than forward simulations; however, we do think that gradual dissemination of findings and contributions (e.g. satellite data exploration) can establish a foundation for subsequent parts. Submitting a forward modeling evaluation paper first will allow us to receive timely feedback and suggestions that can inform any necessary improvements or modifications that need to be made before delving into further aspects of the backward modeling part (e.g. by identifying the problem with not including the top layer in the integration, which the inversion part would likely just have solved by adjusting the state vector).

A few modeling studies using WRF-GHG and methane observations over Europe have been conducted in recent years (e.g. Zhao et al., 2019, 2022; Gałkowski et al., 2020), most of them using ground-based (e.g. ICOS) and column (e.g. TCCON)

observations. In our manuscript, we evaluate simulated $XCH_4$ concentrations resulted from the coupling of a number of atmospheric models against improved TROPOMI observations (Lorente et al., 2022). This new TROPOMI data set was made publicly to the community during the second half of 2022 which means that its use has not been extensively explored. The domain-wide correlation coefficients (0.4–0.5) and root-mean-square errors (27–30 ppb) that we obtained for $XCH_4$ (taking into account the smoothing effect) using this new product are in line with previous studies conducted over specific locations in Central Europe (e.g. Zhao et al., 2019, 2022; Tsuruta et al., 2023).

The second paper will not only focus on the backward component but will also include a first methane emission estimates for Denmark and subsequent comparison against cloud-based products such as the Integrated Methane Inversion (IMI) v1.0 (Varon et al., 2022). As implementing inversion systems based on satellite platforms requires a lot of work and time, publishing the system's core components in companion papers allows a convenient way to disseminate such systems throughout the process.

Gałkowski, M. et al.: Estimating emissions of methane and carbon dioxide sources using analytical Bayesian inversion system based on WRF-GHG tagged tracer simulations, EGU General Assembly, doi:10.5194/egusphere-egu2020-16082, 2020.

Lorente, A. et al.: Evaluation of the methane full-physics retrieval applied to TROPOMI ocean sun glint measurements, Atmos. Meas. Tech., 15, 6585-6603, 2022.

Tsuruta, A. et al.: $CH_4$ Fluxes Derived from Assimilation of TROPOMI $XCH_4$ in CarbonTracker Europe-$CH_4$: Evaluation of Seasonality and Spatial Distribution

in the Northern High Latitudes, Remote Sensing, 15, 1620, doi:10.3390/rs15061620, 2023.

Varon, D. J. et al.: Integrated Methane Inversion (IMI 1.0): a user-friendly, cloud-based facility for inferring high-resolution methane emissions from TROPOMI satellite observations, Geosci. Model Dev., 15, 5787-5805, 2022.

Zhao, X. et al.: Understanding greenhouse gas (GHG) column concentrations in Munich using WRF, Atmos. Chem. Phys., doi:10.5194/acp-2022-281, 2022.

Zhao, X. et al.: Analysis of total column $CO_2$ and $CH_4$ measurements in Berlin with WRF-GHG, Atmos. Chem. Phys., 19, 11279-11302, 2019.

*L170: Do these "agricultural" fluxes include rice production? This is usually separate and somewhat complex given the overlap with naturally inundated areas.*

Yes, EDGARv6.0 has dedicated special effort in including seasonal profiles for the rice cultivation sector. The recent Rice Atlas produced by the International Rice Research Iinstitute (IRRI) (Laborte et al., 2017), which provides a comprehensive rice calendar with monthly specification at country to sub-country level, is already taken into account by EDGAR.

Laborte, A. G. et al.: Rice Atlas, a spatial database of global rice calendars and production, *Scientific Data*, 4, 170074, doi:10.1038/sdata.2017.74, 2017.

*L213: Is Sitch 2003 the correct reference? It makes no mention of methane nor wetlands… More details are needed here as to how the wetland CH4 fluxes are derived.*

In WRF-GHG, the approach of Sitch et al. (2003) is used to estimate the carbon decomposition rate based on WRF predicted fields of soil moisture and temperature. The carbon decomposition rate is then used to estimate the amount of heterotrophic

respiration. Finally, the methane fluxes from wetlands are determined as a percentage of the heterotrophic respiration following the approaches of Christensen et al. (1996) and Kaplan et al. (2002). Section 2.2.2 Biogenic fluxes was rewritten and now it better describes how $CH_4$ fluxes from wetlands are calculated, lines 215-219 in the new version of the manuscript: "$CH_4$ fluxes from wetlands are determined as a percentage of the heterotrophic respiration (Christensen et al., 1996) using the approach of Sitch et al. (2003) and the WRF-GHG variables soil moisture and soil temperature. A wetland inundation map (Kaplan et al., 2002) is then applied for the determination of the wetland fraction per grid cell. $CH_4$ fluxes from termites…" → "$CH_4$ fluxes from wetlands are based on the wetland model developed by Kaplan (2002). This model is based on a diagnostic approach that determines $CH_4$ emissions from wetlands as a percentage of the heterotrophic respiration following the approach of Christensen et al. (1996). The heterotrophic respiration is previously calculated based on a carbon decomposition rate and WRF-GHG variables soil moisture and soil temperature following the approach of Sitch et al. (2003). $CH_4$ fluxes from termites…"

Kaplan, J. O.: Wetlands at the last Glacial Maximum: Distribution and methane emissions, Geophys. Res. Lett., 29, 1079, 2002.

Sitch, S. et al.: Evaluation of ecosystem dynamics, plant geography and terrestrial carbon cycling in the LPJ dynamic global vegetation model, Global Change Biology, 9, 161-185, 2003.

Christensen, T. et al.: Methane flux from northen wetlands and tundra, Tellus, 48B, 652-661, 1996.

*L220: It would be good to see a map of these biogenic fluxes, comparable to Figure 2. Are they sensible? Has the WRF-GHG soil moisture/temperature been evaluated?*

Unlike anthropogenic emissions which are static fields externally prepared (and read in during simulation), the biogenic fluxes are calculated online based on the approaches of Christensen et al. (1996) and Sanderson (1996) for wetlands and termites, respectively, and on Ridgwell et al. (1999) for soil uptake, the only terrestrial sink of methane. A brief description on how these biogenic fluxes are calculated can be found in section 2.2.2. The description of methane emissions from wetlands was rewritten as suggested in the previous comment, and a map of biogenic fluxes for May 2018 similar to Figure 2 is now included in the Supplement (Figure S1). It was verified that no significant natural wetlands were found over the modeling domain, with termites being the main natural contributor to $CH_4$ emissions over the region, lines 225-228 in the new version of the manuscript: "It was verified that no significant natural wetlands were found over the modeling domain, with termites and soil uptake being the primary sources and sinks of $CH_4$ emissions in the region. Figure S1 in the Supplement shows the temporal mean spatial distribution of $CH_4$ emission rate for natural sources and sinks, averaged over the period from May 1 to 31, 2018."

The WRF-GHG parameters soil moisture and soil temperature have not been evaluated. Model evaluation (and improvement) for this kind of parameters is usually made using observational data from field campaigns.

Christensen, T. et al.: Methane flux from northen wetlands and tundra, Tellus, 48B, 652-661, 1996.

Ridgwell, A. J. et al.: Consumption of atmospheric methane by soils: A process-based model, Global Biochem. Cy., 13, 59-70, 1999.

Sanderson, M. G.: Biomass of termites and their emissions of methane and carbon dioxide: A global database, Global Biochem. Cy., 10, 543-557, 1996.

*L235: This is lacking detail. What were the test permutations for the parameterisations? What was the final configuration?*

**To choose between the diverse physics schemes and initial and boundary conditions for meteorological fields and methane concentrations, eight model test runs were made by permuting the cumulus cloud schemes Grell-Freitas and Kain-Fritsch, the planetary boundary layer schemes MYJ and YSU, and initial and boundary conditions from GFS and ERA5 for meteorological fields and from CAM-chem and CAMS for methane background concentrations. All the other physics (see Table 2.1 in Beck et al. (2011)) and emission schemes remained constant all over the model simulations, with the final configuration being shown in Table 2.**

**Section 2.3 Experimental design was rewritten and now it better describes how the model sensitivity tests were conducted. Also, sentences explaining the criteria for selecting the study periods are included in this section, lines 242-257 in the new version of the manuscript: "Initially, a model sensitivity analysis for evaluating physics schemes such as planetary boundary layer and cumulus clouds, and global forcings for meteorological fields and $CH_4$ concentration, was carried out over several two-week periods in 2018 and 2019. Each of these two-week periods were previously examined to have at least 75% of days with TROPOMI $XCH_4$ data covering large portions of Europe. As a result, the physics schemes Yonsei University (YSU) for planetary boundary layer and Kain-Fritsch for cumulus clouds, together with initial and boundary conditions from the European Centre for Medium-Range Weather Forecasts (ECMWF) Reanalysis v5 (ERA5) model (Hersbach et al., 2020), for meteorological processes, and from the NCAR Community Atmosphere Model with Chemistry (CAM-chem) (Lamarque et al., 2012; Emmons, et al., 2020), for background concentrations of $CH_4$, were selected and then used to perform a one-year simulation period from April**

01, 2018 to March 31, 2019. This period was defined based on the following criteria: i) availability of TROPOMI $XCH_4$ data, ii) latest available year data of EDGARv6.0 emissions for $CH_4$, and iii) no occurrence of sustained and irregular scenarios in terms of emissions (e.g., large-scale fire outbreaks and emission reductions associated with COVID-19 lockdowns). Table 2 lists the physics and emissions schemes used in the simulations, with physics schemes other than planetary boundary layer and cumulus clouds being selected based on Beck et al. (2011). A schematic of the model running process is depicted in Appendix A. Off-line initial and boundary conditions derived from the simulations at 30 km are used as input to feed the simulations at 10 km. Model results and discussion for the nested domain are under development and will be described in a forthcoming paper."

Beck, V. et al.: The WRF Greenhouse Gas Model (WRF-GHG), Technical Report No. 25, Max Planck Institute for Biogeochemistry, Jena, Germany, 2011.

*L338: What are the implications of this regridding? What was the approach taken for the averaging kernel and a priori information? More details are needed.*

As briefly mentioned in the manuscript (lines 525-527), regridding techniques can potentially produce data gaps when applied to sparse data, thus reducing the number of grid points, especially when regridding from high-resolution to low-resolution. In our case, the regridding techniques bilinear, conservative and nearest neighbour (source to destination) were used to regrid the satellite data (5.5km×7km) to the WRF-GHG grid (30km×30km), with all of them producing similar results. The a priori profiles and averaging kernels were also regridded using the three techniques previously mentioned, with the bilinear method being finally selected as it preserves the original fine grid structure best.

The first paragraph in section 2.3.1 Postprocessing was rewritten and now it includes some additional information, lines 259-263 in the new version of the manuscript: "In order to compare the simulated $XCH_4$ concentrations with the observations, a set of model data posprocessing steps involving a priori information from the satellite retrievals were carried out as follows: (i) satellite information for each orbit was regridded to the WRF-GHG discretization; (ii) simulated concentrations were resampled to the SRON S5P-RemoTeC standard twelve-levels pressure grid; (iii) smoothed concentrations corresponding to the resampled profiles were calculated according to the following linear transformation:" → "In order to compare the simulated $XCH_4$ concentrations with the observations, a set of model data posprocessing steps involving the satellite retrievals were carried out as follows: (i) the a priori profiles and averaging kernels for each orbit were regridded to the WRF-GHG discretization using a bilinear interpolation; (ii) the simulated concentrations were resampled to the SRON S5P-RemoTeC standard twelve-levels pressure grid; (iii) the smoothed concentrations corresponding to the resampled profiles were calculated according to the following linear transformation:"

Also, the sentence in lines 345-347 was rewritten as "regridded" here is redundant: "… more fully. For ease of model-satellite data comparison, the satellite data were initially regridded to the model grid and then both satellite and model data were flattened to a one-dimensional array. Overall, as described in section 4, the model simulations of meteorological parameters and methane concentrations were in good..." → "...more fully. To facilitate the statistical evaluation of the model-satellite comparison, both the satellite and model data were transformed into one-dimensional arrays. Subsequently, Equations (4), (5) and (6) were applied to compute domain-wide

statistics. Overall, as described in section 3, the simulated CH$_4$ concentrations were in good..."

Figure 3: Missing units. A single colourbar could be used (it's just repeated).

**Figures 3, 4 and 5 now include the units and also a single colorbar.**

Figure 3: The ICOS data all looks to have similar values throughout the year with very little variability compared to the simulations.

**Yes, exactly. This can be also seen in Figure S1 in the Supplement. The high variability in the simulated concentrations near the surface is strongly tied to the anthropogenic emissions – note that the methane variability from background signals is quite similar to that from ICOS observations. The influence of anthropogenic emissions on methane concentrations diminishes gradually with increasing height, as expected.**

L394: I don't believe this is true (but I could be wrong). Specifically I'm thinking of Tsuruta et al. (2023) who state "Anthropogenic fluxes, such as those from agriculture, landfills and production and use of oil, gas and coal, are taken from the EDGAR v6.0 inventory".

**Yes, Tsuruta et al. (2023) used EDGARv6.0 for anthropogenic fluxes. This work had not been included/cited in the discussion section because it was not available by the time when we submitted our manuscript to GMD. New discussions on Tsuruta et al. (2023) results have now been included in several parts throughout the manuscript, lines 397-401 in the new version of the manuscript: "...anthropogenic sources. No inverse modeling studies of CH$_4$ emissions based entirely on EDGARv6.0 for anthropogenic sources have been conducted over Europe. However, a recent inversion approach for CH$_4$ emissions over China..." ➔ "...anthropogenic sources. Using EDGARv6.0 CH$_4$**

fluxes as the a priori emission estimates and two sets of TROPOMI-based $XCH_4$ observations, the global inversion approach conducted by Tsuruta et al. (2023) showed that over central Europe the anthropogenic $CH_4$ emissions would be slightly overestimated, mainly during spring and autumn. However, higher emission estimates are otherwise found when ground-based data is used to drive the inversions. The inversion approach for $CH_4$ emissions over China…"

And lines 598-605: "$XCH_4$ signals from natural sources (wetlands and termites) and biomass burning were not relevant during the study period. According to Kaplan (2002), potential natural wetlands in the 30 km modeling domain concentrate over the Baltic countries, Belarus and western regions of Russia. Among the factors that could have negatively influenced the accumulation of biospheric $CH_4$ in the atmosphere over the study region are: a less $CH_4$ formation tied to the extremely dry season in summer 2018 over central and northern Europe (Rousi et al., 2022); a $CH_4$ compensation by soil uptake processes; and transport mechanisms. Yu et al. (2022)..." ➔ "$XCH_4$ signals from natural sources (wetlands and termites) and biomass burning were not relevant during the study period. The inversion estimates for 2018 conducted by Tsuruta et al. (2023) showed that, compared to the anthropogenic emissions, the wetland emissions over central Europe were small, mainly during summer months when biogenic fluxes reached their minimum values. Among the factors that could have negatively influenced the accumulation of biospheric $CH_4$ in the atmosphere over the study region are: a less $CH_4$ formation tied to the extremely dry season in summer 2018 over central and northern Europe (Rousi et al., 2022); a $CH_4$ compensation by soil uptake processes as the fluxes are dominated by mineral soils which are mostly net sink of $CH_4$ (Tsuruta et al., 2023); and transport mechanisms. According to the Kaplan wetland map, potential

natural wetlands in Europe concentrate over western regions of Russia. Yu et al. (2022)…"

Kaplan, J. O.: Wetlands at the last Glacial Maximum: Distribution and methane emissions, Geophys. Res. Lett., 29, 1079, 2002.

Rousi, E. et al.: The extremely hot and dry 2018 summer in central and northern Europe from a multi-faceted weather and climate perspective, EGUsphere, doi:10.5194/egusphere-2022-813, 2022.

Tsuruta, A. et al.: $CH_4$ Fluxes Derived from Assimilation of TROPOMI $XCH_4$ in CarbonTracker Europe-$CH_4$: Evaluation of Seasonality and Spatial Distribution in the Northern High Latitudes, Remote Sensing, 15, 1620, doi:10.3390/rs15061620, 2023.

Yu, X. et al.,: A high-resolution satellite-based map of global methane emissions reveals missing wetland, fossil fuel and monsoon sources, EGUsphere, doi:105194/egusphere-2022-948, 2022.

*L495: This comes back to why this time period specifically was selected and also the point earlier about how some of the fluxes were calculated (e.g. wetland emissions).*

Section 2.3 Experimental design was rewritten and now it includes the criteria for selecting the study periods for both the model sensitivity tests and the one-year period from April 01, 2018 to March 31, 2019 (see comment L98's response). Also, section 2.2.2 Biogenic fluxes was rewritten and now it better describes how $CH_4$ fluxes from wetlands are calculated (see comment L213's response).

*L519: The modelled stratosphere can play a significant role and I don't see a mention of that. Has any attempt been made to assess how well the modelled stratosphere performs (e.g. by comparison to profile observations or other sources)?*

**As this first part aims to evaluate the WRF-GHG model for a one-year simulation period, special focus has been given to platforms that continuously measure methane concentrations such as ICOS and TROPOMI. Model evaluation of methane vertical profiles in the stratosphere is usually performed using observations from meteorological balloons, spectrometers and aircrafts. Aircraft observations are especially suited for studying the troposphere-stratosphere exchange as they regularly reach high altitudes.**

**For the one-year study period from April 01, 2018 to March 31, 2019, methane data from meteorological balloons and COCCON spectrometers as those used by Tsuruta et al. (2023) and Tu et al. (2020), respectively, are available for most of 2018 over the European Arctic region, out of the AUMIA modeling domain. On the other hand, a total of approximately 55 h of high-frequency aircraft observations of methane between May and June 2018 were obtained aboard HALO in the scope of the CoMet 1.0 campaign. Observations were performed at altitudes ranging from 50 m up to 14 km above mean sea level (Gałkowski et al., 2021). This lack of sufficient vertical information hampers to perform any model evaluation of the stratospheric methane, with most of the methane modeling studies using space observations to evaluate their total column (troposphere+stratosphere) concentrations.**

**Gałkowski, M. et al.: In situ observations of greenhouse gases over Europe during the CoMet 1.0 campaign aboard the HALO aircraft, Atmos. Meas. Tech., 14, 1525-1544, 2021.**

**Tsuruta, A. et al.: $CH_4$ Fluxes Derived from Assimilation of TROPOMI $XCH_4$ in CarbonTracker Europe-$CH_4$: Evaluation of Seasonality and Spatial Distribution**

in the Northern High Latitudes, Remote Sensing, 15, 1620, doi:10.3390/rs15061620, 2023.

Tu, Q. et al.: Atmospheric CO2 and CH4 abundances on regional scales in boreal areas using CAMS reanalysis, COCCON spectrometers and Sentinel-5 Precursor satellite observations, Atmospheric Measurement Techniques, 13, 4751-4771, 2020.

*Figure 6 – Caption: I think the "respectively" needs to be moved outside of the brackets as I think it also applies to panels a/e.*

**The "respectively" in Figure 6 caption only applies to panels b and f which represent XCH₄ fields with and without smoothing. Both the panels a and e represent the same temporal mean spatial distributions of TROPOMI XCH₄ concentrations over the study period.**

*Figure 6: Panels C/G – This difference is dominated by the offset and there's very little spatial structure visisble (i.e. it's all light red or all light blue). It may be more informative to centre the colourbar around the average and lessen the range to enahcne spatial details.*

**Given that we want to quantify how much higher or lower the two sets of model concentrations (with and without smoothing) are with regard to the same satellite estimates, we think that the choice of maps of relative (or absolute, e.g. see Figs 5 and 6 in Tsuruta et al. 2023) differences with the colormap centered on zero facilitates the quantitative analysis and provides a much more consistent baseline for the comparison. A colormap centering on the average is also a good option although simultaneous visualization of different data imbalances would be more difficult to interpretate.**

CH₄ Fluxes Derived from Assimilation of TROPOMI XCH₄ in CarbonTracker Europe-CH₄: Evaluation of Seasonality and Spatial Distribution in the Northern High Latitudes
Tsuruta, A. et al.
2023
Remote Sensing
10.3390/rs15061620
15
en

**Tsuruta, A. et al.: CH$_4$ Fluxes Derived from Assimilation of TROPOMI XCH$_4$ in CarbonTracker Europe-CH$_4$: Evaluation of Seasonality and Spatial Distribution in the Northern High Latitudes, Remote Sensing, 15, 1620, doi:10.3390/rs15061620, 2023.**

**Referee #2**

*General comment:*

*The manuscript describes the first part of the AUMIA system, which focuses on the forward modelling with WRF-GHG and its evaluation using TROPOMI and ICOS observations. The major concern is that without the inverse modelling part of the work, this first paper does not include much of a model development but focuses on forward modelling evaluation. In addition, there are several methodological descriptions missing, that should be clarified, that I listed below. Other than these aspects, the manuscript is well-written and easy to follow and understand. However, before being suitable for publishing in GMD, the below comments need to be addressed and implemented.*

**We understand the reviewer's concern that without the inverse modeling part not much of a model development has been exhibited so far, especially considering that the inversion part by itself captures much more attention than forward simulations; however, we do think that gradual dissemination of findings and contributions can establish a foundation for subsequent parts. Submitting a forward modeling evaluation paper first will allow us to receive timely feedback and suggestions that can inform any necessary improvements or modifications that need to be made before delving into further aspects of the backward modeling part (e.g. by identifying the problem with not including the top layer in the integration, which the inversion part would likely just**

have solved by adjusting the state vector). Additional methodological descriptions on the study period definition, model sensitivity tests and how emission fluxes are converted into atmospheric concentrations, have now been included. Also, a better model description on how $CH_4$ emissions from biogenic sources are calculated, has been accomplished (Referee #1 suggestion).

*Specific comments:*

*Lines 34-36. This last sentence sounds like an overstatement as there are previous studies using TROPOMI observations.*

The first part of the sentence "The results found in this study contribute with a new model evaluation of methane concentrations over Europe" is true in the sense that new EDGAR emission estimates (Ferrario et al., 2021) together with improved TROPOMI observations (Lorente et al., 2022) were used to evaluate the methane concentrations over Europe. Regarding the second part "and demonstrate a huge and under explored potential for methane inverse modeling using improved TROPOMI products in large-scale applications", it can be supported by the domain-wide statistical metrics which are in line with previous studies conducted over specific locations in Central Europe (e.g. Zhao et al., 2019, 2022; Tsuruta et al., 2023). The more satellite observations are improved, the more accurate the inversion estimates tend to be. The term "and under explored" has been removed to not insinuate the TROPOMI data we used here have been available for so long and that even so it's been under explored, lines 34-36 in the new version of the manuscript: "The results found in this study contribute with a new model evaluation of methane concentrations over Europe, and demonstrate a huge and under explored potential for methane inverse modeling using improved TROPOMI products in large-scale applications." ➔ "The results found in this study contribute with

a new model evaluation of methane concentrations over Europe, and demonstrate a huge potential for methane inverse modeling using improved TROPOMI products in large-scale applications."

**Also in lines 652-654:** "The results found in this study are in line with previous studies conducted over urban areas in central Europe, and thus, demonstrate a huge and under explored potential for $CH_4$ inverse modeling using updated TROPOMI $XCH_4$ data sets in large-scale applications." ➔ "The results found in this study are in line with previous studies conducted over urban areas in central Europe, and thus, demonstrate a huge potential for $CH_4$ inverse modeling using updated TROPOMI $XCH_4$ data sets in large-scale applications."

**Also, the proper EDGARv6.0 reference is now included in the manuscript, lines 174-175:** "Anthropogenic fluxes of $CH_4$ (not including biomass burning sources) are externally prepared based on the Emissions Database for Global Atmospheric Research (EDGAR) version 6 Greenhouse Gas Emissions (Crippa et al., 2021). " ➔ "Anthropogenic fluxes of $CH_4$ (not including biomass burning sources) are externally prepared based on the Emissions Database for Global Atmospheric Research (EDGAR) version 6 Greenhouse Gas Emissions (Ferrario et al., 2021). "

Ferrario, M. et al.: EDGAR v6.0 Greenhouse Gas Emissions. European Commision, Joint Research Centre (JCR) [Dataset] PID: http://data.europa.eu/89h/97a67d67-c62e-4826-b873-9d972c4f670b, 2021.

Lorente, A. et al.: Evaluation of the methane full-physics retrieval applied to TROPOMI ocean sun glint measurements, Atmos. Meas. Tech., 15, 6585-6603, 2022.

Tsuruta, A. et al.: $CH_4$ Fluxes Derived from Assimilation of TROPOMI $XCH_4$ in CarbonTracker Europe-$CH_4$: Evaluation of Seasonality and Spatial Distribution

in the Northern High Latitudes, Remote Sensing, 15, 1620, doi:10.3390/rs15061620, 2023.

Zhao, X. et al.: Understanding greenhouse gas (GHG) column concentrations in Munich using WRF, Atmos. Chem. Phys., doi:10.5194/acp-2022-281, 2022.

Zhao, X. et al.: Analysis of total column $CO_2$ and $CH_4$ measurements in Berlin with WRF-GHG, Atmos. Chem. Phys., 19, 11279-11302, 2019.

*Line 98-99: How are these periods selected? This should be better described.*

The main criteria for selecting the two-week periods for model sensitivity tests was to have at least 75% of days with TROPOMI data covering large portions of Europe. Large numbers of observation/model pairs (that spread out across the modeling domain) allow to perform a more representative domain-wide statistical evaluation. The one-year period from April 01, 2018 to March 31, 2019 was selected because of the following reasons: 1) availability of TROPOMI operational data and the improved TROPOMI data from March 2018 onwards; 2) evaluate the most recent EDGARv6.0 emissions for methane (2018); and 3) avoid sustained irregular scenarios in terms of emissions, e.g., fire outbreaks in most part of 2019 and emission reductions associated with COVID-19 lockdowns in 2020 and 2021, both at global scale. The 2018 summer was particularly interesting to focus on because of the larger than average number of cloud-free days over Denmark and most part of Europe.

Section 2.3 Experimental design was rewritten and now it includes the criteria for selecting the study periods for the model sensitivity tests, lines 242-250 in the new version of the manuscript: "Initially, a model sensitivity analysis for evaluation of model parameterizations such as planetary boundary layer and cumulus clouds, as well as global forcings for $CH_4$ concentration, was carried out over several two-week periods in

**2018 and 2019. Then, based on the model configuration that best fit the satellite data, one-year simulation period from...”** ➔ **“Initially, a model sensitivity analysis for evaluating physics schemes such as planetary boundary layer and cumulus clouds, and global forcings for meteorological fields and CH$_4$ concentration, was carried out over several two-week periods in 2018 and 2019. Each of these two-week periods were previously examined to have at least 75% of days with TROPOMI XCH$_4$ data covering large portions of Europe. As a result, the physics schemes Yonsei University (YSU) for planetary boundary layer and Kain-Fritsch for cumulus clouds, together with initial and boundary conditions from the European Centre for Medium-Range Weather Forecasts (ECMWF) Reanalysis v5 (ERA5) model (Hersbach et al., 2020), for meteorological processes, and from the NCAR Community Atmosphere Model with Chemistry (CAM-chem) (Lamarque et al., 2012; Emmons, et al., 2020), for background concentrations of CH$_4$, were selected and then used to perform a one-year simulation period from…”**

**And for the one-year simulation period, lines 250-253:** **“This period was defined based on the following criteria: i) availability of TROPOMI XCH$_4$ data, ii) latest available year data of EDGARv6.0 emissions for CH$_4$, and iii) no occurrence of sustained and irregular scenarios in terms of emissions (e.g., large-scale fire outbreaks and emission reductions associated with COVID-19 lockdowns).”**

*Line 119: What are these flux models and how do they work? More information is needed here.*

**Basically, since each gas occupies the same volume under the same atmospheric pressure and temperature, all gas species can be converted from mol/km$^2$/h to Δ[ppmv] (response to changes in pressure and temperature) using the same approach.**

**Mathematically, the two-dimensional flux variable of a gas specie (emis_ant) is multiplied by a conversion factor (conv_rho) and then added to the first layer of the three-dimensional tracer variable of that gas specie (chem).**

**chem = chem + conv_rho\*emis_ant**

**conv_rho = 8.0461e-6\*(1/rho_phy\*dtstep/dz8w)**

**where rho_phy, dtstep and dz8w denote the air density [kg/m$^3$], model time step [s] and the thickness of the first model layer [m], respectively. 8.0461e-6 is the molar mass of air per second [g/mol/s]. The file module_ghg_fluxes.F in /chem contains all subroutines for adding the emissions of $CH_4$, $CO_2$, and CO calculated per time step to the corresponding atmospheric concentrations. An additional sentence on how emission fluxes are converted into atmospheric concentrations is now included in the manuscript, lines 121-125 in the new version of the manuscript:** **"...are converted into atmospheric concentrations based on flux models. On the other hand, online calculations comprise $CH_4$ emissions from wetlands and termites, and $CH_4$ uptake by soil. $CH_4$ contributions from anthropogenic..."** ➔ **"...are converted into atmospheric concentrations based on an incremental approach. The $CH_4$ concentration changes are calculated as the $CH_4$ emission multiplied by a conversion factor that depends on the air density and thickness of the first model layer. On the other hand, $CH_4$ fluxes from wetlands and termites, as well as $CH_4$ uptake by soil, are all calculated online in the simulations (see section 2.2.2 for further details). $CH_4$ contributions from anthropogenic..."**

*Table 1 can be considered to be moved to the supplement.*

As Table 1 shows attributes that are not described in section 2.1.1 Grid configuration, we do think it should be kept on the manuscript.

*Lines 235-237: More information is needed for these sensitivity simulations.*

To choose between the diverse physics schemes and initial and boundary conditions for meteorological fields and methane concentrations, eight model test runs were made by permuting the cumulus cloud schemes Grell-Freitas and Kain-Fritsch, the planetary boundary layer schemes MYJ and YSU, and initial and boundary conditions from GFS and ERA5 for meteorological fields and from CAM-chem and CAMS for methane background concentrations. All the other physics (see Table 2.1 in Beck et al. (2011)) and emission schemes remained constant all over the model simulations, with the final configuration being shown in Table 2.

Section 2.3 Experimental design was rewritten and now it better describes how the model sensitivity tests were conducted. Also, sentences explaining the criteria for selecting the study periods are included in this section, lines 242-257 in the new version of the manuscript: "Initially, a model sensitivity analysis for evaluating physics schemes such as planetary boundary layer and cumulus clouds, and global forcings for meteorological fields and $CH_4$ concentration, was carried out over several two-week periods in 2018 and 2019. Each of these two-week periods were previously examined to have at least 75% of days with TROPOMI $XCH_4$ data covering large portions of Europe. As a result, the physics schemes Yonsei University (YSU) for planetary boundary layer and Kain-Fritsch for cumulus clouds, together with initial and boundary conditions from the European Centre for Medium-Range Weather Forecasts (ECMWF) Reanalysis v5 (ERA5) model (Hersbach et al., 2020), for meteorological processes, and from the NCAR Community Atmosphere Model with Chemistry (CAM-

chem) (Lamarque et al., 2012; Emmons, et al., 2020), for background concentrations of $CH_4$, were selected and then used to perform a one-year simulation period from April 01, 2018 to March 31, 2019. This period was defined based on the following criteria: i) availability of TROPOMI $XCH_4$ data, ii) latest available year data of EDGARv6.0 emissions for $CH_4$, and iii) no occurrence of sustained and irregular scenarios in terms of emissions (e.g., large-scale fire outbreaks and emission reductions associated with COVID-19 lockdowns). Table 2 lists the physics and emissions schemes used in the simulations, with physics schemes other than planetary boundary layer and cumulus clouds being selected based on Beck et al. (2011). A schematic of the model running process is depicted in Appendix A. Off-line initial and boundary conditions derived from the simulations at 30 km are used as input to feed the simulations at 10 km. Model results and discussion for the nested domain are under development and will be described in a forthcoming paper."

Beck, V. et al.: The WRF Greenhouse Gas Model (WRF-GHG), Technical Report No.
25, Max Planck Institute for Biogeochemistry, Jena, Germany, 2011.

*Lines 244-250: This section and Table 2 are identical, just keep one of them.*

**Section 2.3 Experimental design was rewritten and Table 2 kept.**

*ICOS stations in Figures 3-5 seem to not change, should be double-checked.*

**Yes, the $CH_4$ concentrations from ICOS stations did not change significantly near the surface (0 to 100 m) during the study period. This can be also observed in Figure S1 in the Supplement, with $CH_4$ concentrations ranging roughly from 1970 to 2030 ppb.**

*Editorial comments:*

*Line 15: Remove "a" before powerful tools.*

**Corrected.**

*Line 28: Remove "otherwise" and add "On the other hand" in the beginning of the sentence.*

**Corrected. A part of the sentence has been rewritten, lines 28-29 in the new version of the manuscript: "…respectively. For winter months, otherwise, model-observation discrepancies show a significant..." ➜ "...respectively. On the other hand, model-observation discrepancies for winter months show a significant..."**

*Line 39: Add a reference after the first sentence.*

**A reference has been included to support the sentence, line 39: "Atmospheric methane ($CH_4$) has more than doubled since the pre-industrial. Although it..." ➜ "Atmospheric methane ($CH_4$) has more than doubled since the pre-industrial era (Meinshausen et al., 2017). Although it..."**

**Meinshausen, M. et al.: Historical greenhouse gas concentrations for climate modelling (CMIP6), Geosci. Model Dev., 10(5), 2057-2116, 2017.**

*Figures 3-5. Units are missing in the figures and/or the figure caption.*

**Figures 3, 4 and 5 and their captions now include the units and also a single colorbar.**